# ALMTokenizer: A Low-bitrate and Semantic-rich Audio Codec Tokenizer for Audio Language Modeling

Dongchao Yang [1]   Songxiang Liu [2]   Haohan Guo [1]   Jiankun Zhao [1]   Yuanyuan Wang [1]   Helin Wang [2]   Zeqian Ju [2]   Xubo Liu [2]   Xueyuan Chen [1]   Xu Tan [2]   Xixin Wu [1]   Helen Meng [1]

## Abstract

Recent advancements in audio language models have underscored the pivotal role of audio tokenization, which converts audio signals into discrete tokens, thereby facilitating the application of language model architectures to the audio domain. In this study, we introduce ALMTokenizer, a novel low-bitrate and semantically rich audio codec tokenizer for audio language models. Prior methods, such as Encodec, typically encode individual audio frames into discrete tokens without considering the use of context information across frames. Unlike these methods, we introduce a novel query-based compression strategy to capture holistic information with a set of learnable query tokens by explicitly modeling the context information across frames. This design not only enables the codec model to capture more semantic information but also encodes the audio signal with fewer token sequences. Additionally, to enhance the semantic information in audio codec models, we introduce the following: (1) A masked autoencoder (MAE) loss, (2) Vector quantization based on semantic priors, and (3) An autoregressive (AR) prediction loss. As a result, ALMTokenizer achieves competitive reconstruction performance relative to state-of-the-art approaches while operating at a lower bitrate. Within the same audio language model framework, ALMTokenizer outperforms previous tokenizers in audio understanding and generation tasks.[1]

[1]The Chinese University of Hong Kong, Hong Kong, China [2]Independent Researchers. Correspondence to: Dongchao Yang <dcyang@se.cuhk.edu.hk>, Helen Meng <hmmeng@se.cuhk.edu.hk>.

[1]http://dongchaoyang.top/ALMTokenizer/

## 1. Introduction

The field of generative modeling has witnessed remarkable progress, largely driven by the success of autoregressive (AR) models in the development of large language models (LLMs) (OpenAI, 2023). Inspired by the success of LLMs in the fields of natural language processing (NLP), recent works have begun to employ AR transformers for audio generation (Borsos et al., 2023a; Agostinelli et al., 2023; Yang et al., 2023c), such as using the AR transformer paradigm to solve text-to-speech task (Wang et al., 2023), or expanding the text LLM into multimodal LLM by integrating the audio modality into the original LLM (Défossez et al., 2024). Audio tokenizer plays an important role in all of these models, which converts audio signals into discrete token sequence for AR audio language modeling.

In the literature, audio codec models, such as SoundStream (Zeghidour et al., 2021) and Encodec (Défossez et al., 2022), have been widely adopted as audio tokenizers for audio language models. These generative models aim to represent audio data in a quantized discrete latent space, where the codec's decoder is then used to reconstruct the audio signals from the generated discrete token sequences. Recently, there has been significant interest in the audio community regarding audio codec tokenizers, leading to the proposal of several novel models (Kumar et al., 2023; Ji et al., 2024; Défossez et al., 2024; Parker et al., 2024; Zhang et al., 2023). Despite the advancements in audio codec models, an important research question remains unanswered: **which type of audio codec is most suitable for audio language modeling?** Inspired by previous works (Borsos et al., 2023a; Parker et al., 2024; Ji et al., 2024; Défossez et al., 2024), these studies investigate two key properties of audio codec models: low bitrate and semantic richness. We first conduct a set of evaluation experiments to explore the influence of bitrate and semantic information on audio language modeling. Specifically, we train three audio codec models with varying bitrates, while keeping the number of vector quantization (VQ) layers constant and adjusting the frame rates to 50 Hz, 25 Hz, and 12.5 Hz. We then train the audio language model using different audio tokenizers on the same dataset. To assess the impact of semantic information, we

also train a 12.5 Hz semantic tokenizer and incorporate it into the audio language model. Further details can be found in Appendix B. Figure 1 presents the results, which show that: (1) low-bitrate audio codec models significantly enhance training and inference efficiency; and (2) semantic information is more easily modeled by LM-based generative methods, *e.g.* lower PPL and loss. The experimental findings demonstrate the importance of constructing a low-bitrate and semantic-rich audio codec tokenizer for audio language modeling. Based on these results, we propose a novel audio codec tokenizer that offers the following advantages: (1) Low-bitrate: it compresses the audio data into fewer tokens; (2) Semantic-rich: it incorporates abundant semantic information; (3) AR-driven latent space: it optimizes the latent space for autoregressive (AR) modeling.

To achieve this objective, we propose the following novel techniques: (1) We introduce a novel query-based compression strategy, which uses a set of learnable query tokens to capture holistic information by explicitly modeling the context information across audio frames with transformer layers. This strategy effectively takes advantage of the strong modeling capabilities of transformers to achieve better compression and semantic modeling. It also enables dynamic control over the compression rate by adjusting the number of query tokens. (2) To enhance semantic richness in the codec model, we introduce a Masked Autoencoder (MAE) loss, which encourages the model to capture more global information. (3) Inspired by previous works (Zhu et al., 2024), we propose the integration of semantic priors into the VQ layer. Specifically, we perform k-means clustering on the pre-trained wav2vec2 (Baevski et al., 2020) and BEATs (Chen et al., 2022b) encoder outputs, using the cluster centers to initialize the VQ layer. (4) We observe that AR models struggle to fit the distribution of the residuals in the VQ layers, with token prediction accuracy being notably lower in the second and third VQ layers compared to the first. To address this issue, we introduce an AR prediction loss to optimize the latent space.

To evaluate the effectiveness of the ALMTokenizer, we first compare its reconstruction and semantic performance with previous state-of-the-art models. Using the same audio language model framework, we then demonstrate that ALMTokenizer achieves superior performance in LM-based audio understanding and generation tasks, including text-to-speech (TTS), speech-to-text (ASR), audio captioning, text-to-sound, text-to-music, and music captioning.

## 2. Related Works

### 2.1. Audio Language Models

Recently, there has been a growing interest in bridging audio and text through multimodal learning approaches. Models such as AudioLM (Borsos et al., 2023a) leverage AR trans-

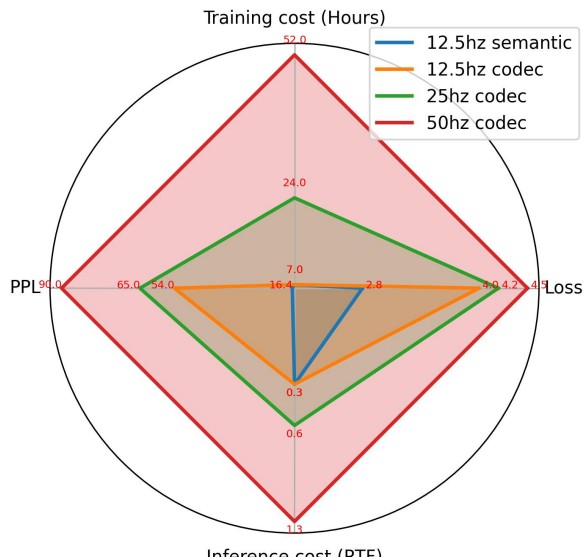

*Figure 1.* The performance comparison when different types of tokenizer is used for audio modeling. PPL refers to perplexity.

formers and hierarchical modeling techniques to process audio data directly, learning representations that capture both linguistic and acoustic features. Inspired by AudioLM, VALL-E (Wang et al., 2023) and SPEAR-TTS (Kharitonov et al., 2023) formulate the text-to-speech task as an audio language modeling problem: generating an audio token sequence with the help of an autoregressive transformer. MusicLM (Agostinelli et al., 2023) and MusicGen (Copet et al., 2023) frame the text-to-music task as an audio language modeling problem. UniSep (Wang et al., 2025) explores using audio LM to solve audio separation tasks with the help of audio tokenizer. Moshi (Défossez et al., 2024), SpiRit-LM (Nguyen et al., 2025), and GLM4-Voice (Zeng et al., 2024) explore speech-to-speech conversation. Furthermore, audio tokenizers can also be combined with discrete diffusion models (Yang et al., 2023d;a; Borsos et al., 2023b; Ju et al., 2024).In all of these models, the audio tokenizer plays a crucial role by transforming audio data into a discrete latent sequence, reducing computational demands compared to directly processing the audio signal, and enhancing the effectiveness and efficiency of the generation process.

### 2.2. Audio Tokenizer

In the literature, both semantic and acoustic tokenizers are widely employed in audio language models. The semantic tokenizer is trained using pre-trained self-supervised learning (SSL) models, such as Hubert (Hsu et al., 2021) and WavLM (Chen et al., 2022a). Applying k-means or vector quantization in these models generates semantic tokens (Zeng et al., 2024; Du et al., 2024; Liu et al., 2024). Previous works (Borsos et al., 2023a) demonstrate that semantic tokens are more easily modeled by language models. However, due to the loss of significant acoustic information

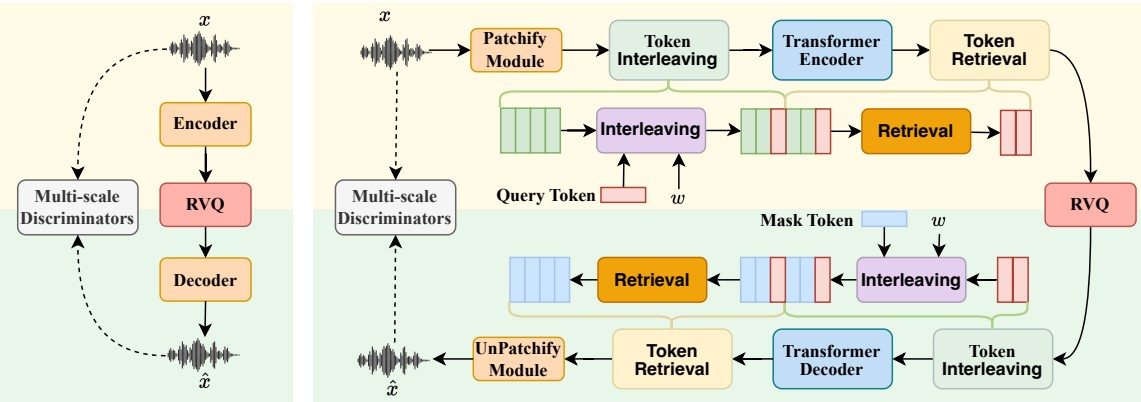

*Figure 2.* The left part illustrates the framework of the previous audio codec, while the right part provides an overview of the proposed ALMTokenizer. $w$ denotes the window size. The details of ALMTokenizer can be found in Section 3.2.

in semantic tokens, they rely on an additional decoder to generate high-fidelity waveform, such as a diffusion model (Ho et al., 2020) or flow-matching (Lipman et al., 2022). Inevitably, this additional module results in increased inference complexity and poorer reconstruction.

Acoustic tokenizer refers to audio codec models, trained for acoustic-level reconstruction tasks. Audio codecs (Zeghidour et al., 2021; Défossez et al., 2022; Yang et al., 2023b; Kumar et al., 2023) have demonstrated exceptional performance in reconstructing high-quality audio. In general, these codec models consist of an encoder, a quantizer, and a decoder. Both the encoder and decoder are lightweight, resulting in minimal inference costs. Compared to semantic tokens, codec models can support audio, speech, and music domains, and their rich acoustic details mitigate the need for cascading architectures in downstream generative models. Recently, an increasing number of audio codec models have been proposed, focusing on (1) Better reconstruction quality, such as DAC (Kumar et al., 2023), Vocos (Siuzdak, 2023), SQ-Codec (Yang et al., 2024c;b) and APCodec (Ai et al., 2024); (2) Low-bitrate models, such as HiFiCodec (Yang et al., 2023b), wavtokenizer (Ji et al., 2024), StableCodec (Parker et al., 2024), and TS3-Codec (Wu et al., 2024); (3) Task-driven codecs, designed for text-to-speech tasks, such as FACodec (Ju et al., 2024), SpeechTokenizer (Zhang et al., 2023), Single-Codec (Li et al., 2024), audio retrieval-based Tokenizers (Banerjee & Arora, 2022; van Niekerk et al., 2024). In this study, we focus on developing a low-bitrate, semantically rich audio codec tokenizer. The most closely related work to ours is MimiCodec (Défossez et al., 2024), which provides high-quality semantic information while achieving a low bitrate (1.1 kbps). However, MimiCodec relies on knowledge distillation from WavLM (Chen et al., 2022a) to the first VQ layer, whereas the remaining VQ layers do not incorporate semantic information. Furthermore, it is specifically designed for speech tasks and has not been

validated for non-speech tasks, such as sound and music generation. In contrast to MimiCodec, our ALMTokenizer encodes more semantic information across all VQ layers, achieves a lower bitrate, and is designed for both speech and general sound.

## 3. Proposed Method

This section introduces the technical details of the proposed ALMTokenizer. Section 3.1 presents the framework of previous audio codec models. Section 3.2 presents the details of proposed audio codec framework. In Sections 3.3 and 3.4, we present the training loss and training strategies.

### 3.1. Preliminary

Previous audio codecs (Défossez et al., 2022; Zeghidour et al., 2021) typically adopt an encoder-quantizer-decoder framework, as shown in the left part of Figure 2. The audio is encoded into several audio frames by the encoder. Then, residual vector quantization (RVQ) (Zeghidour et al., 2021) is used to quantize these audio frames. Lastly, the decoder is used to recover the waveform from the quantized audio frames. It can be observed that previous works treat each audio frame equally and rely on these quantized frames to recover the audio. However, such a strategy (1) ignores the fact that different audio frames encode different levels of information, which results in some audio frames being difficult to recover in low-bitrate settings (e.g., encoding the audio frames at 12.5 Hz); (2) fails to utilize the context information between different frames.

### 3.2. Query-based Audio Compression

To construct a low-bitrate, semantically rich audio codec model, we propose a query-based compression strategy. Our

approach is inspired by the success of MAE (He et al., 2022), which applies a masking operation to the original image with a high mask rate (75%). With the help of a transformer encoder and decoder, it is possible to recover the masked image content by utilizing the context information between different patches. Thus, we propose using a group of query tokens [2] to capture holistic audio context information from the audio frames with the assistance of a transformer encoder. Since these query tokens include rich context information, it is possible to reconstruct the audio based on them. Then, a transformer decoder and mask tokens are employed to reconstruct the audio from the quantized query tokens. This strategy leverages the powerful modeling capabilities of transformers to achieve better compression and semantic modeling. Similar query-based strategies has been widely explored in previous works, such as BLIP2 (Li et al., 2023), SALMONN (Tang et al., 2024) and TiTok(Yu et al., 2024). The right part of Figure 2 illustrates the overall framework of ALMTokenizer. In the following sections, we detail each component and the associated training loss.

**Patchify and UnPatchify** We explore two types of Patchify modules: (1) Following Encodec (Défossez et al., 2022), a convolution-based module, which encodes the audio data $x$ into $e \in \mathcal{R}^{T \times d}$, where $T$ and $d$ denote the number of frames and the vector dimension, and (2) Following StableCodec (Parker et al., 2024), which directly uses a linear layer to encode the audio data into $e \in \mathcal{R}^{T \times d}$ and adds several transformer layers. Similarly, the UnPatchify mirrors the architecture of Patchify. If we use the Encodec-style Patchify module, the UnPatchify module substitutes stride convolutions with transposed convolutions and reverses the stride order. If we use the StableCodec-style Patchify module, the UnPatchify module includes a transformer block and a reshape operation. In our preliminary experiments, we find that the Encodec-style Patchify and UnPatchify modules bring better reconstruction performance. We adopt the Encodec-style Patchify module as our default setting.

**Token Interleaving** The token interleaving module aims to combine two token sequences into a single sequence. In the encoder part, we combine the audio frames $e \in \mathcal{R}^{T \times d}$ and the query token [CLS]. Assuming a window size of $w$, the query token will be inserted into the audio frame sequence at every $w$-intervals. In the decoder part, the token interleaving module is used to combine the quantized query tokens and learnable mask tokens. We insert $w$ mask tokens before each query token. During the training stage, we dynamically choose the window size for each training iteration.

**Token Retrieval** The token retrieval module aims to retrieve the relevant tokens from a sequence. In the encoder part, we

[2]Query tokens are learnable embedding vectors that are updated throughout the training process.

use it to retrieve the learnable query tokens. In the decoder part, we use it to retrieve the learnable mask tokens.

**Query-based Transformer Encoder** As the previous part discussed, we introduce a learnable query token [cls] $\in \mathcal{R}^{1 \times d}$ to capture holistic information from the audio frames $e$. As Figure 2 shows, we first combine the audio frames and query token using a token interleaving module with a window size $w$. Then, a transformer module is applied to model the whole sequence $e_a$. After that, we employ a token retrieval module to extract the query tokens $h \in \mathcal{R}^{\lfloor T/w \rfloor \times d}$.

$$e = P(x), e_a = Interleaving(e, cls, w),$$
$$e_a = En(e_a), h = Rectrieval(e_a, w) \quad (1)$$

where $P(\cdot)$ denotes the Patchify module. $En(\cdot)$ denotes the transformer encoder.

**Residual Vector Quantization** To build a low-bitrate audio codec, we empirically set the number of RVQ layers to 3, since we found that 3 RVQ layers suffice to build an effective audio codec model: $\hat{h} = Q(h)$. Inspired by previous works (Zhu et al., 2024; Yang et al., 2024a), we first obtain the k-means clusters of Wav2vec2 (Baevski et al., 2020) to represent the speech semantic prior, and the k-means clusters of the BEATs (Chen et al., 2022b) to represent the general sound semantic prior. Assuming the codebook size is $C$, we set $C/2$ to represent speech, with the remaining portion representing general sound. We then use these semantic priors to initialize the codebook of the VQ layer and fix it. Next, we apply a linear layer to map the input features into the VQ layer.

**Query-based Transformer Decoder** To recover the audio information, we construct a reverse process using the encoder part. We first use the token interleaving module to combine the mask token $m \in \mathcal{R}^{1 \times d}$ with $\hat{h}$. The new sequence is then modeled by a transformer module. We expect that these mask tokens can be used to recover the audio information with the help of the Unpatchify module.

$$q_a = Interleaving(\hat{h}, m, w), q_a = De(q_a)$$
$$e_o = Rectrieval(q_a, w), \hat{x} = UnP(e_o), \quad (2)$$

where $Unp(\cdot)$ denotes the Unpatchify module. $De(\cdot)$ denotes the transformer decoder.

### 3.3. Training Loss

Similar to previous audio codecs, our approach is based on a GAN objective, where we optimize both the generator (which consists of the Patchify module, transformer encoder, quantizer, transformer decoder, and UnPatchify module) and the discriminators. For the generator, the training loss comprises four components: (1) reconstruction loss term; (2) adversarial loss term; (3) Masked AutoEncoder (MAE) loss; and (4) AR prediction loss. The reconstruction and

adversarial losses typically follow previous works (Défossez et al., 2022; Zeghidour et al., 2021). In the following, we describe the MAE loss and AR prediction loss. More details of training loss refer to Appendix G.

**MAE Loss** As we discussed in Section 1, a semantic-rich audio codec tokenizer is better suited for audio language modeling. Inspired by the success of MAE (He et al., 2022), we propose to incorporate an MAE loss during the training of the audio codec. Specifically, for the frame sequence $e$, we randomly choose several audio frame features and set these frames to zero, $e_m = \text{Mask}(e)$. We pass the masked features $e_m$ into the encoder transformer. Then, the encoded features are passed into an MAE-decoder transformer block to predict $e$. In our experiments, we adopt a dynamic mask rate (from 0.2 to 0.3), we found that using a large mask rate will significantly influence the reconstruction performance. Following MAE (He et al., 2022), we apply the MSE loss to the masked audio frames.

**AR Loss** As shown in figure 3, we find that the first layer of RVQ-based audio codec models is easier to fit for the audio language model than the other layers (e.g., layers 2 and 3). One possible reason is that the first layer encodes more semantically related information. For speech data, most of the content information can be recovered by the first VQ layer, while the residual layers primarily encode acoustic-level information, which influences speech quality. To make the tokens in the residual layer easier to fit, we introduce an autoregressive (AR) prediction prior (Wang et al., 2024a) in the RVQ latent space. Specifically, we introduce a lightweight continuous autoregressive (AR) transformer [3], which is used to conduct next-token prediction in the RVQ layer. For example, it is tasked with predicting the quantized feature of the third VQ layer based on the features of the first and second VQ layers. We use mean squared error (MSE) loss for optimization.

$$p_\theta = \prod_{i=1}^{3} p_\theta(\boldsymbol{x}_i | \boldsymbol{x}_1, ..., \boldsymbol{x}_{i-1}, \theta) \tag{3}$$

where $\theta$ denotes the parameter of AR transformer.

### 3.4. Two-stage Training Strategy

Although training the ALMTokenizer using the typical Encodec (Défossez et al., 2022) setting is feasible, we introduce a two-stage training paradigm to improve both reconstruction performance and semantic information. Our motivation stems from the fact that audio codec quantization focuses on modeling local relationships, whereas seman-

---

tic information focuses on modeling global relationships. These two goals are in conflict. To resolve this conflict, we present a two-stage training strategy. In the first stage, we do not incorporate the quantization part; instead, we train directly an AutoEncoder with Patchify and UnPatchify modules. To encode more semantic information in the Patchify module, we introduce MAE loss during this stage, by adding transformer-based MAE-encoder and decoder. The encoder processes the masked frame sequence, and the decoder predicts the masked part. After training, the transformer encoder and decoder are discarded. In the second stage, we first initialize the ALMTokenizer's Patchify and UnPatchify modules with the checkpoint from the first stage, and freeze the parameters of the Patchify module. Then, we train the model using the training loss described in Section 3.3.

## 4. Experiments

### 4.1. Dataset and Training Details

**Data preparation for the audio codec** ALMTokenizer is trained on approximately 4,500 hours of data. In the speech domain, we utilize LibriTTS training set (Zen et al., 2019) and a subset of Multilingual LibriSpeech (MLS) (Pratap et al., 2020), with 2,000 hours randomly selected. In the sound domain, we utilize a subset of AudioSet, with 1,000 hours randomly selected; in the music domain, we employ a subset of the Million Song Dataset (Bertin-Mahieux et al., 2011), also with 1,000 hours randomly selected. We evaluate the codec's speech reconstruction performance using a subset of the VCTK dataset (Veaux et al., 2017), and assess both audio and music reconstruction performance using the AudioCaps (Kim et al., 2019) validation set and the MusicCaps dataset (Agostinelli et al., 2023), respectively.

**Data for Audio Language Models** To assess the effectiveness of the proposed audio tokenizer, we construct an audio language model framework to perform six audio-related tasks. The details are provided in Appendix D.3 and D.4. For speech data, we select 2,000 hours of speech-text pairs from LibriHeavy (Kang et al., 2024). For sound data, we utilize the AudioCaps training set and BBC Sound Effects. For music data, we use a subset of the Million Song dataset and the caption data from LP-MusicCaps (Doh et al., 2023).

**Implementation Details** ALMTokenizer first performs patchification on the audio data, we set the patch size to 320 in all of experiments, which encodes 1 second of 24kHz audio into 75 frames. For the Encodec-style Patchify module, we adopt the settings from Encodec (Défossez et al., 2022) encoder. To enable streaming for the codec model, a causal convolution layer is employed. For the encoder-transformer and decoder-transformer components, we use 24 self-attention layers, with latent dimensions of 256 and 512, respectively. Following StableCodec (Parker et al.,

---

[3]The term continuous autoregressive (AR) transformer is used to distinguish our approach from traditional discrete AR models, which operate on discrete token sequences and are optimized using cross-entropy loss. In our study, to facilitate gradient backpropagation, we apply the AR transformer directly to continuous features.

*Table 1.* The speech reconstruction and semantic performance comparison between the ALMTokenizer and previous tokenizers. FPS denotes that the frame number in one second. TPS denotes that the token number in one second. CS denotes the codebook size, BR denotes the bit-rate. ST denotes speechtokenizer. **Bold** for the best result and underline for the second-best result. Evaluation on VCTK dataset.

| Models | FPS/TPS | CS/BR | Reconstruction | | | | | Semantic | |
| | | | UTMOS (↑) | DNS-MOS (↑) | VISQOL (↑) | STOI (↑) | PESQ (↑) | ASR (↓) | ER (↑) |
|---|---|---|---|---|---|---|---|---|---|
| Hubert (Hsu et al., 2021) | - | - | - | - | - | - | - | 6.5 | 31.0 |
| WavLM (Chen et al., 2022a) | - | - | - | - | - | - | - | 6.2 | 29.0 |
| Encodec (Défossez et al., 2022) | 50/150 | 1024/1.5kbps | 2.58 | 3.27 | 3.64 | 0.81 | 2.0 | 35.3 | 26.5 |
| DAC (Kumar et al., 2023) | 50/150 | 1024/1.5kbps | 3.13 | 3.41 | 3.67 | 0.81 | **2.1** | 44.1 | 17.6 |
| Wavtokenizer (Ji et al., 2024) | 40/40 | 4096/0.48kbps | 3.67 | 3.50 | 3.72 | 0.79 | 1.9 | 44.6 | 19.8 |
| StableCodec (Parker et al., 2024) | 25/25 | 46656/0.4kbps | **4.22** | **3.64** | 3.40 | 0.76 | 1.8 | 98.3 | 15.8 |
| ST (Zhang et al., 2023) | 50/150 | 1024/1.5kbps | 3.41 | 3.36 | 3.68 | 0.79 | 1.7 | 19.8 | 27.0 |
| Mimi (Défossez et al., 2024) | 12.5/37.5 | 2048/0.41kbps | 3.01 | 3.14 | 3.28 | 0.75 | 1.5 | 25.1 | 28.0 |
| Mimi (Défossez et al., 2024) | 12.5/100 | 2048/1.1kbps | 3.65 | 3.38 | **3.82** | **0.82** | **2.1** | 23.8 | 28.3 |
| ALMTokenizer (Ours) | 12.5/37.5 | 2048/0.41kbps | 3.76 | **3.64** | 3.78 | 0.81 | 2.0 | **18.3** | **29.0** |

2024), the self-attention mechanism uses a causal sliding attention window of 64 steps to restrict the receptive field and promote the generalization of the architecture to sequences of arbitrary length. Rotary Positional Embeddings (RoPE) are used. Refer to Appendix G for the details of ALMTokenizer model training. For the audio language model, we follow the framework of Moshi (Défossez et al., 2024). For further details, refer to Appendix A.

## 4.2. Evaluation Metrics

We evaluate the performance of previous SOTA audio tokenizers, and our proposed ALMTokenizer across audio reconstruction, audio semantic information, audio understanding, and audio generation tasks.

**Audio Reconstruction** For speech reconstruction, we use DNS-MOS, UT-MOS, PESQ, STOI (Short-time Objective Intelligibility), and VISQOL. For sound and music data evaluation, VISQOL (audio version), STFT loss, and Mel loss are used. Furthermore, following (Kumar et al., 2023), the MUSHRA subjective test is conducted for speech, sound, and music. Refer to Appendix D for more details.

**Audio Semantic Information** Previous SSL models, such as Hubert (Hsu et al., 2021), have shown that semantic-rich representation can be used to solve downstream recognition tasks by fine-tuning several adaptor layers. Thus, we can validate the performance of features of the audio tokenizer for downstream recognition tasks. For speech data, we conduct the automatic speech recognition (ASR) task on the LibriSpeech (Panayotov et al., 2015) dataset, and the emotion classification (EC) task on the EMOVO (Costantini et al., 2014) dataset. For sound data, we conduct sound classification tasks on the ESC-50 dataset (Piczak, 2015). For music data, we conduct music classification tasks on the Medley-solos-DB dataset (Lostanlen & Cella, 2016).

**Audio Understanding** To further validate whether the audio

*Table 2.* The sound reconstruction peformance comparison between the proposed ALMTokenizer and previous audio tokenizer models. SC denotes the sound classification task. Evaluation on AudioCaps validation set.

| Models | ViSQOL (↑) | Mel loss (↓) | STFT loss (↓) | SC (↑) |
|---|---|---|---|---|
| BEATs | - | - | - | 24% |
| Wav2vec2 | - | - | - | 53% |
| Encodec | **3.05** | 16.3 | **1.23** | 15% |
| DAC | 2.98 | 17.6 | 1.24 | 20% |
| Wavtokenizer | 2.18 | 32.7 | 2.50 | 12% |
| Ours | 2.99 | **15.0** | 1.24 | **44%** |

*Table 3.* The music reconstruction and semantic performance comparison between the ALMTokenizer and previous audio tokenizers. MC denotes the music classification task. Evaluation on Musicaps dataset.

| Models | ViSQOL (↑) | Mel loss (↓) | STFT loss (↓) | MC (↑) |
|---|---|---|---|---|
| BEATs | - | - | - | 54% |
| Wav2vec2 | - | - | - | 65% |
| Encodec | 4.04 | 34.8 | **1.26** | 45% |
| DAC | **4.06** | 35.9 | 1.28 | 48% |
| Wavtokenizer | 3.85 | 48.2 | 1.47 | 54% |
| Ours | 3.96 | **34.4** | 1.32 | **59%** |

tokenizer is suitable for building an audio language model, we propose to conduct an understanding task using discrete tokens. We conduct three tasks: ASR, audio caption, and music caption. For the audio data, we use the audio tokenizer to transform it into discrete tokens, and for text data, we use the BPE tokenizer of LLAMA 3.2. For audio and music caption, we follow (Drossos et al., 2020) and adopt BLEU-1, BLEU-2, BLEU-3, METEOR, ROUGE-L, CIDEr-D, SPICE, and SPIDEr metrics.

**Audio Generation** We also conduct audio generation tasks, including text-to-speech, text-to-sound, and text-to-music. Refer to Appendix D for more details.

## 4.3. The Reconstruction and Semantic Performance

We first compare the reconstruction and semantic performance of ALMTokenizer with previous audio tokenizers. Table 1 presents the speech reconstruction and semantic results. We observe the following: (1) In terms of reconstruction, ALMTokenizer achieves impressive results in the low-bitrate setting. For example, compared with previous SOTA models, MimiCodec and Wavtokenizer, ALMTokenizer achieves better reconstruction performance at a lower bitrate. We also note that StableCodec performs well on UT-MOS. The main reason is that StableCodec has denoising capabilities, while the original audio includes some noise. This explains why StableCodec achieves good results on UT-MOS but performs poorly on PESQ and STOI. (2) In terms of semantic information, ALMTokenizer demonstrates superior performance, e.g, ALMTokenizer outperforms previous SOTA models, such as Wavtokenizer and StableCodec [4]. Notably, in the emotion classification task, ALMTokenizer achieves performance comparable to previous SSL models, such as Hubert and WavLM. However, we also note that ALMTokenizer still lags behind these SSL models in ASR performance. We speculate that the inclusion of acoustic information may detract from ASR performance, despite ALMTokenizer containing rich semantic information. Table 2 and 3 show the sound and music experimental results. We can see that ALMTokenizer demonstrates strong reconstruction performance under the low-bitrate setting. Compared to WavTokenizer, the reconstruction performance shows significant improvement. Furthermore, we also note that sound and music are inherently more complex than speech, and encoding them at very low-bitrate remains a challenge. In terms of semantic information, ALMTokenizer significantly surpasses previous works, such as WavTokenizer and Encodec. In comparison with SSL models, BEATs (Chen et al., 2022b) and Wav2vec2-audioset version, ALMTokenizer shows comparable performance. We also perform the MUSHRA subjective test for the reconstruction performance. As shown in Table 7, we find that ALMTokenizer effectively maintains strong subjective reconstruction performance on speech, music, and audio, even with a very low-bitrate setting.

## 4.4. Audio Understanding and Generation Results

**Speech Understanding and Generation Tasks** Table 4 shows the LM-based TTS and ASR results. For the TTS task, we mainly focus on robustness and speech quality. In terms of robustness, we can see that the GLM4-voice tokenizer (Zeng et al., 2024), MimiCodec, and the proposed ALMTokenizer bring better performance than others, highlighting the importance of semantic information

---

[4]StableCodec's feature dimension is 6, it is hard to apply it for down-streaming task by simple fine-tuning

---

*Table 4.* The LM-based TTS and ASR results. The first three metrics are used for TTS, while the last one is used for ASR. GLM4-Voice (Zeng et al., 2024) is a single layer semantic tokenizer. Evaluation on LibriSpeech test clean set.

| Models | WER (↓) | DNSMOS (↑) | UT-MOS (↑) | ASR (↓) |
|---|---|---|---|---|
| GLM4-voice | 9.9 | 3.96 | 3.79 | 16.3 ± 1.5 |
| DAC | 24.5 | 3.14 | 2.06 | 58.4 ± 1.2 |
| Encodec | 22.9 | 3.48 | 2.14 | 77.2 ± 2.3 |
| StableCodec | 22.7 | 3.63 | 3.70 | 28.0 ± 1.9 |
| Wavtokenizer | 18.5 | 3.72 | 3.58 | 45.6 ± 2.7 |
| MimiCodec | 16.0 | 3.67 | 2.93 | 23.1 ± 1.5 |
| **Ours** | **11.7** | **3.75** | **3.88** | **19.6 ± 1.8** |

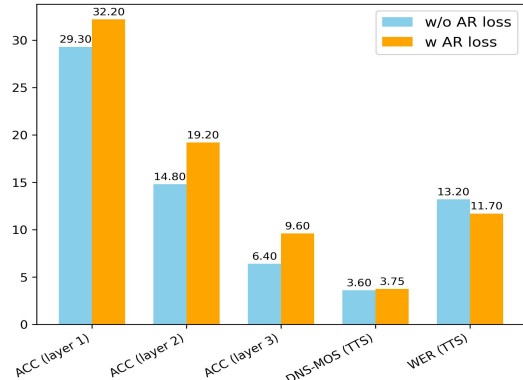

*Figure 3.* The performance comparison with or without AR loss.

for LM-based speech generation. Compared to previous audio codec tokenizers, ALMTokenizer brings significant improvement. In terms of generated speech quality, ALMTokenizer also shows great advantages, further demonstrating that the proposed tokenizer is more suitable for audio language modeling. Similarly, when we conduct the ASR task using discrete tokens as input, semantic information is also important. Traditional audio codec models perform poorly in this setting, such as DAC, Encodec, and WavTokenizer. StableCodec was fine-tuned by using a CTC head to predict the force-aligned phoneme tags from pre-bottleneck latents. MimiCodec distills the semantic information from WavLM. Thus, they have better performance than previous codec models. In ALMTokenizer, we propose a novel codec framework and training loss to better encode semantic information in the codec model.

**Sound/music Understanding and Generation Results** We conduct text-to-sound, text-to-music, audio caption and music caption tasks within the same audio language model framework. The experimental results shown in Table 5 indicate that ALMTokenizer shows better performance in both audio caption and audio generation tasks, further demonstrating its advantages. We put more audio tokenizer reconstruction performance experiments on Appendix F, including evaluation on LibriTTS test set, length generalization,

*Table 5.* The LM-based sound, music understanding and generation. B1, B2, B3, RG, ME, CD, SP, and SD denote BLEU-1, BLEU-2, BLEU-3, METEOR, ROUGE-L, CIDEr-D, SPICE, and SPIDEr, respectively. Evaluation on Audiocaps and Musicaps datasets.

| Models | Understanding | | | | | | | | Generation | | |
|---|---|---|---|---|---|---|---|---|---|---|---|
| | B1 (↑) | B2(↑) | B3 (↑) | ME (↑) | RG (↑) | CD (↑) | SP (↑) | SD (↑) | FD (↓) | FAD (↓) | KL (↓) |
| **Sound Task** | | | | | | | | | | | |
| Encodec | 0.25 | 0.15 | 0.08 | 0.11 | 0.24 | 0.57 | 0.14 | 0.35 | 10.03 | 8.22 | 1.73 |
| DAC | 0.26 | 0.15 | 0.08 | 0.11 | **0.26** | 0.51 | 0.13 | 0.32 | 14.14 | 11.7 | 1.55 |
| Wavtokenizer | 0.24 | 0.14 | 0.08 | 0.10 | 0.22 | 0.38 | 0.11 | 0.25 | 6.76 | **4.55** | 1.28 |
| **ALMTokenizer (Ours)** | **0.28** | **0.17** | **0.11** | **0.12** | 0.24 | **0.60** | **0.15** | **0.37** | **4.11** | 6.16 | **0.55** |
| **Music Task** | | | | | | | | | | | |
| Encodec | 0.30 | 0.14 | **0.08** | 0.11 | 0.23 | 0.37 | 0.09 | 0.23 | 7.22 | 5.48 | 1.06 |
| DAC | 0.29 | 0.14 | 0.08 | 0.11 | 0.23 | 0.37 | 0.09 | 0.23 | 12.89 | 8.36 | 1.68 |
| Wavtokenizer | 0.19 | 0.06 | 0.02 | 0.06 | 0.13 | 0.06 | 0.05 | 0.05 | 4.39 | 11.93 | 0.88 |
| **ALMTokenizer (Ours)** | **0.34** | **0.15** | 0.07 | **0.13** | **0.25** | **0.44** | **0.10** | **0.27** | **3.55** | **4.58** | **0.43** |

and compared to diffusion-based audio codec models.

## 4.5. Ablation Study

In order to gain a more comprehensive understanding of ALMTokenizer, we systematically compared each key component using a controlled experimental setup, employing identical architectures and hyperparameters across all trials. **The Effectiveness of Query-based Audio Compression** In this study, we propose a query-based audio compression strategy for compressing audio data in a very low-bitrate setting. To validate its effectiveness, we follow previous audio codec models, such as MimiCodec (Défossez et al., 2024). In the encoder part, we use a stride size of [8, 6, 5, 4, 2] to compress 1-second, 24 kHz audio into 12.5Hz, followed by applying 3 RVQ layers to quantize it. As shown in Table 6, using previous audio codec frameworks makes it difficult to maintain good reconstruction performance in very low-bitrate settings. As a result, the proposed query-based compression method is more effective in this setting. **The Influence of Semantic Prior for VQ** To explore the influence of semantic priors on the audio codec model, we conduct an experiment where we remove the semantic prior and instead train a learnable RVQ following Encodec. As shown in Table 6, we find that updating the RVQ layer improves reconstruction performance but reduces semantic information, demonstrating that integrating semantic priors into the VQ layer enhances semantic information. **The Influence of MAE Loss** We also conduct experiments to evaluate the effectiveness of the MAE loss. As shown in Table 6, we find that the MAE loss is crucial for enhancing the semantic information in the codec model. Although the MAE loss has a slight negative effect on reconstruction, it is a crucial factor in building a better audio tokenizer. **The Influence of AR Loss** From Table 6, we observe that adding the AR loss reduces reconstruction performance. In Figure 3, we compare token prediction accuracy and TTS performance with and without LM loss. We observe that using LM loss significantly improves token prediction accuracy, particularly for the second and third VQ layers, which

shows the effectiveness of our motivation and solution. **The Influence of Two-stage Training** As Table 6 shows, the two-stage training strategy is crucial as it significantly improves reconstruction performance and semantic information in the codec model. **The Influence of Patchify Module** We investigate two types of Patchify modules: Encodec-style and StableCodec-style. As shown in Table 6, using Encodec-style Patchify modules yields better performance. One possible reason is that StableCodec-style Patchify modules (Parker et al., 2024) may depend on larger data and model parameters, as the original paper scales their model to 1B. In contrast, we use only four transformer layers to ensure a fair comparison with Encodec-style modules. Due to page limitations, we defer the ablation study on the influence of window size $w$ in query-based compression, codebook size, the influence of mask-rate, and model size on reconstruction to Appendix C.

## 4.6. Discussion

In this section, we discuss two fundamental questions in audio tokenization. **Question 1: Is a single quantization layer better than multiple quantization layers? Question 2: Does a low-bit rate with high reconstruction performance define a good audio tokenizer?**
**Question 1** Although WavTokenizer and StableCodec demonstrate the potential to build a low-bitrate audio codec tokenizer with a single quantization layer, they rely on a higher frame rate (e.g., 25 or 40 Hz). As shown in Figure 1, a lower frame rate (e.g., 12.5 Hz) is critical for improving training efficiency. Thanks to UniAudio (Yang et al., 2023c) and Moshi's (Défossez et al., 2024) audio language model framework, multiple quantization layers do not increase the sequence length. Therefore, multiple quantization layers present an effective approach for building a low-bitrate, semantically rich audio codec.
**Question 2** To address this question, we present two comparisons. First, as shown in Tables 4 and 1, StableCodec exhibits better reconstruction performance and a lower bit-rate compared to WavTokenizer. However, when applied to the

*Table 6.* Ablation study of codec framework, training loss, and training strategy. ASR and ER are used to evaluate the semantic information. The others are used to evaluate the reconstruction performance. Experiments conducts on VCTK dataset.

| Setting | UTMOS (↑) | DNSMOS (↑) | VISQOL (↑) | PESQ (↑) | STOI (↑) | ASR (↓) | ER (↑) |
|---|---|---|---|---|---|---|---|
| ALMTokenizer | 3.76 | 3.64 | 3.78 | 2.0 | 0.81 | 18.3 | 29.0 |
| **Framework ablation** | | | | | | | |
| w/o the query-based framework | 2.49 | 3.13 | 3.37 | 1.58 | 0.77 | 34.5 | 22.6 |
| Only query-based framework | 3.54 | 3.41 | 3.44 | 1.69 | 0.78 | 27.2 | 24.5 |
| **Training loss ablation** | | | | | | | |
| w/o semantic prior for VQ | 3.79 | 3.66 | 3.78 | 2.12 | 0.83 | 19.2 | 28.4 |
| w/o MAE loss | 3.70 | 3.76 | 3.83 | 2.10 | 0.82 | 24.5 | 23.2 |
| w/o AR loss | 3.72 | 3.81 | 3.80 | 2.08 | 0.82 | 18.8 | 30.2 |
| **Different Patchify module** | | | | | | | |
| use Linear-Patchify | 3.47 | 3.36 | 3.27 | 1.78 | 0.78 | 20.3 | 26.7 |
| **Training strategy ablation** | | | | | | | |
| w/o two-stage training | 3.60 | 3.39 | 3.24 | 1.55 | 0.74 | 22.8 | 25.9 |

*Table 7.* The subjective reconstruction results using MUSHRA (comparative scoring of samples) of codec models on speech, sound and music. **Bold** for the best result and underline for the second-best result.

| Models | FPS/TPS | CS/BR | Speech (↑) | Sound (↑) | Music (↑) |
|---|---|---|---|---|---|
| **Speech** | | | | | |
| MimiCodec (3 RVQ) (Défossez et al., 2024) | 12.5/37.5 | 2048/0.41kbps | 65.61 ± 5.2 | - | - |
| MimiCodec (8 RVQ) (Défossez et al., 2024) | 12.5/100 | 2048/1.1kbps | **86.7 ± 2.3** | - | - |
| StableCodec (Parker et al., 2024) | 25/25 | 46656/0.4kbps | 81.7 ± 4.4 | - | - |
| SpeechTokenizer (Zhang et al., 2023) | 50/150 | 1024/1.5bps | 73.7 ± 4.6 | - | - |
| **Audio** | | | | | |
| Encodec (Défossez et al., 2022) | 50/150 | 1024/1.5bps | 75.1 ± 3.9 | **77.2 ± 4.2** | **73.7 ± 4.6** |
| DAC (Kumar et al., 2023) | 50/150 | 1024/1.5bps | 79.3 ± 4.2 | 71.3 ± 4.1 | 71.3 ± 4.1 |
| Wavtokenizer (Défossez et al., 2022) | 40/40 | 4096/0.48bps | 84.0 ± 2.1 | 63.1 ± 4.6 | 54.1 ± 5.4 |
| Ours | 12.5/37.5 | 2048/0.41kbps | 84.8 ± 3.7 | 72.4 ± 4.7 | 69.0 ± 4.5 |

text-to-speech generation task, WavTokenizer demonstrates better robustness. One possible reason for this is that StableCodec uses a large-scale codebook size (46,656), which may increase the modeling complexity. Second, although MimiCodec has a higher bit-rate and poorer reconstruction performance than StableCodec, it demonstrates more stable TTS generation performance and better ASR performance. This phenomenon further underscores the importance of semantic information. In summary, a good audio tokenizer for an audio language model should not only consider low-bitrate and reconstruction, but also account for the semantic information in the codec model.

## 5. Conclusion

In this study, we present a low-bitrate, semantically rich audio codec tokenizer. Specifically, we propose a query-based compression strategy to effectively compress the audio data into a low-bitrate format while incorporating more semantic information. Furthermore, we introduce several training losses to enhance semantic information, including MAE loss and AR loss. Extensive experiments demonstrate the

effectiveness of ALMTokenizer. Within the same audio language modeling framework, ALMTokenizer exhibits superior performance in both understanding and generation tasks. We discuss the limitation of this study in Appendix I.

## Impact Statement

This paper presents an audio tokenizer for audio language models, which can be applied to various audio generation tasks, such as text-to-speech and text-to-music. There is potential for misuse in generating misinformation, deepfake audio, or other harmful content. We advocate for the development of a detection model to identify audio produced by the codec model and generated by other generative models.

## Acknowledgements

This study was supported in part by the Centre for Perceptual and Interactive Intelligence (CPII) Ltd., a CUHK-led InnoCentre under the InnoHK initiative of the Innovation and Technology Commission of the Hong Kong Special Administrative Region Government.

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

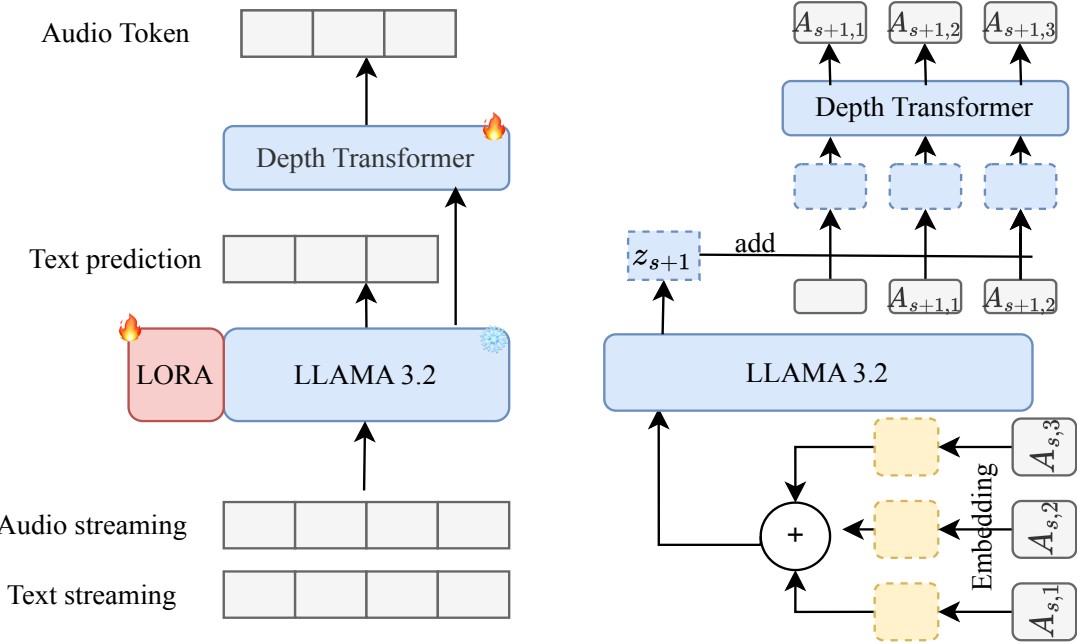

*Figure 4.* The left diagram illustrates the framework of the audio language model, which includes a pre-trained LLM, a LoRA module, and a depth transformer. The audio language model can process both text and audio streaming inputs and generate corresponding text and audio outputs. The right diagram provides details of hierarchical audio modeling.

# A. The details of audio language model framework

In this section, we provide details of the audio language model. We follow the framework of UniAudio (Yang et al., 2023c) and Moshi (Défossez et al., 2024), which combines a pre-trained LLM with a smaller Transformer model to predict audio tokens in a hierarchical manner. In their original paper, both the LLM and the small Transformer are updated during the training process. Due to resource limitations, and following (Hao et al., 2023), we incorporate LoRA (Hu et al., 2021) into the LLM model. For the LLM model, we use the LLAMA3.2 1B version. During training, we update only the LoRA module and the small Transformer.

**LORA setting** For the LoRA module, we add LoRA parameters to the self-attention and linear layers. We set $lora_r = 32$ and $lora_{alpha} = 16$.

**Depth Transformer setting** For the depth transformer, we use 6 self-attention layer. We set the attention head number as 32. The attention dimension is the same as the LLAMA 3.2 1B.

# B. The details of the influence of bitrate and semantic information for audio language model.

In this section, we provide details of the validation experiments to explore the influence of bitrate and semantic information on audio language models. Following AudioLM (Borsos et al., 2023a), we construct an audio token pre-training task similar to text pre-training, where the model is tasked with predicting the next audio token based on the previous token sequence.

### B.1. Training data

We conduct the experiments on 2000 hours speech data, these data is selected from MLS dataset (Pratap et al., 2020).

### B.2. Test data

We evaluate on LibriSpeech test clean set.

*Table 8.* The reconstruction performance of different frame rate of audio tokenizers.

| Version | Bitrate ($\downarrow$) | FPS ($\downarrow$) | codebook size | PESQ ($\uparrow$) | UT-MOS ($\uparrow$) | VISQOL ($\uparrow$) | STOI ($\uparrow$) |
|---|---|---|---|---|---|---|---|
| 50hz | 1650bps | 50 | 2048 | 2.22 | 3.69 | 3.63 | 0.86 |
| 25hz | 825bps | 25 | 2048 | 2.07 | 3.56 | 3.61 | 0.83 |
| 12.5hz | 412.5bps | 12.5 | 2048 | 1.58 | 2.49 | 3.37 | 0.77 |

### B.3. Framework

We use the same framework as described in Section A; the difference is that we do not use text streaming.

### B.4. Three Types of Audio Tokenizers

Following the structure of MimiCodec (Défossez et al., 2024), we train three versions of the audio codec tokenizer. All of the audio codec models are trained on 24kHz speech data. We train three versions of the audio codec models, as follows:

(V1) We set the down-sampling rate to [2, 5, 6, 8], resulting in a 50 Hz frame rate. We use three RVQ layers, and the codebook size is 2,048. The bitrate of this audio codec is 1.65 kbps.

(V2) We set the down-sampling rate to [4, 5, 6, 8], resulting in a 25 Hz frame rate. We use three RVQ layers, and the codebook size is 2,048. The bitrate of this audio codec is 825 bps.

(V3) We set the down-sampling rate to [2, 4, 5, 6, 8], resulting in a 12.5 Hz frame rate. We use three RVQ layers, and the codebook size is 2,048. The bitrate of this audio codec is 412.5 bps.

Note that the original MimiCodec is trained with distillation loss from WavLM; we do not add this loss during the training of our audio tokenizer. Therefore, these three audio tokenizers do not include any semantic information. Table 8 shows the reconstruction performance of the three audio tokenizers.

### B.5. Semantic Tokenizer

The previous three audio codec tokenizers do not consider semantic information. To evaluate the importance of semantic information, we follow WhisperSpeech[5] to build a Whisper-based semantic tokenizer. Specifically, we follow the training code of WhisperSpeech, using two down-sampling layers to compress the Whisper encoder's features into a 12.5 Hz frame rate, and then we add three RVQ layers to quantize them. Thus, this semantic tokenizer has the same bitrate as the V3 audio tokenizer.

### B.6. Evaluation metrics

We evaluate the pre-training performance from the following aspects:

**Training efficiency**: As is well known, the space complexity of a transformer is $O(T^2)$, where $T$ is the sequence length. A low-bitrate audio tokenizer can compress the audio signal into a few token sequences, thereby improving training efficiency. For all experiments, we use the same GPU machine to train the model and record the statistical training duration.

**Inference efficiency**: Similarly, a low-bitrate audio tokenizer can improve inference efficiency, as it requires fewer inference steps. We use the Real-Time Factor (RTF) to assess inference efficiency. Note that for all experiments, we do not use any inference optimization tricks, such as KV cache.

**Validation loss and perplexity**: Following text LLMs (OpenAI, 2023), we use validation loss and perplexity to evaluate model performance.

---

[5]https://github.com/WhisperSpeech/WhisperSpeech

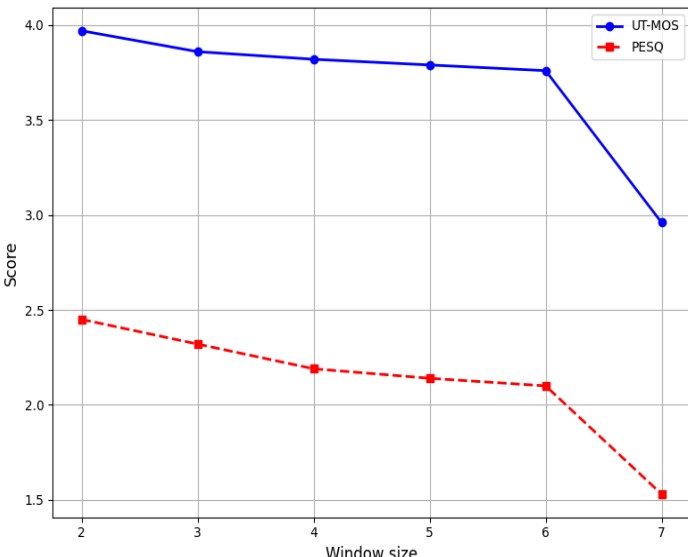

*Figure 5.* The performance comparison with different window size during inference.

*Table 9.* The influence of codebook size for reconstruction performance.

| Codebook Size | PESQ (↑) | UT-MOS (↑) | VISQOL (↑) | STOI (↑) | STFT loss (↓) | Token untilzation (↑) |
|---|---|---|---|---|---|---|
| 2048 | 2.0 | 3.76 | 3.78 | 0.81 | 1.20 | 100% |
| 1024 | 1.83 | 3.66 | 3.65 | 0.80 | 1.14 | 100% |
| 512 | 1.69 | 3.64 | 3.58 | 0.792 | 1.18 | 100% |

## C. Ablation study

### C.1. The influence of window size for ALMTokenizer

As discussed in the previous section, the proposed ALMTokenizer supports a dynamic compression rate by changing the window size $w$. Figure 5 shows the comparison of reconstruction performance with different window sizes. We observe that using a smaller window size results in better reconstruction performance, but it also increases the bitrate. For exmaple, if the window size is 2, the bitrate is 1237.5bps, window size is 6, the bitrate is 412.5. It also shows the advantages of proposed method: we can dynamically change the frame rate during the inference by setting different window size.

### C.2. The influence of codebook size

We explore three different codebook sizes: 512, 1024, and 2048. To align with the setting of MimiCodec (Défossez et al., 2024), we set the max codebook size as 2048. The results, as shown in Table 9, are presented. We observe that scaling the codebook size improves reconstruction performance. Furthermore, we also find that almost all tokens have been used.

### C.3. The influence of model size for reconstruction performance

To explore the influence of model size on reconstruction performance, we set up two configurations: (1) We use 24 self-attention layers for both the transformer encoder and transformer decoder, resulting in 174M parameters. (2) We use 12 self-attention layers for both the transformer encoder and transformer decoder, resulting in 87M parameters. In both settings, we keep the Patchify module the same size, as it consists of several convolutional layers, and its total parameters are small. The experimental results, as shown in Table 10, indicate that using a larger model can improve reconstruction but also increases computational resource consumption (higher RTF). Previous work, StableCodec (Parker et al., 2024), shows that scaling the codec model to 1B parameters can lead to better performance. Due to computational resource limitations, we leave scaling to a larger model size for future work.

Table 10. The influence of model for reconstruction performance.

| Setting | PESQ (↑) | UT-MOS (↑) | VISQOL (↑) | STOI (↑) | Model size (↓) | RTF (↓) |
|---|---|---|---|---|---|---|
| 24 attention layer | **2.0** | **3.76** | **3.78** | **0.81** | 174 | 0.031 |
| 12 attention layer | 1.87 | 3.57 | 3.70 | 0.79 | 87 | 0.019 |

## C.4. The influence of mask-rate in MAE loss

Inspired by MAE(He et al., 2022), we tested three group of mask rates ranges: (10–20%), (20–30%), and (30–40%). The experiments as following Table shows. Results indicate that higher rates (30–40%) benefit semantics but harm reconstruction, leading us to adopt an intermediate range (20–30%).

Table 11. The influence of mask-rate for MAE loss.

| mask rate range | UTMOS | DNSMOS | VISQOL | PESQ | STOI | ASR | ER |
|---|---|---|---|---|---|---|---|
| 10-20% | 3.77 | 3.62 | 3.80 | 2.0 | 0.81 | 18.7 | 27.7 |
| 20-30% | 3.76 | 3.64 | 3.78 | 2.0 | 0.81 | 18.3 | 29.0 |
| 30-40% | 3.36 | 3.06 | 3.31 | 1.58 | 0.77 | 18.1 | 29.6 |

# D. Evaluation

We evaluate the performance of the previous SOTA audio tokenizers and our proposed ALMTokenizer across audio reconstruction, audio semantic information, audio understanding, and audio generation tasks.

## D.1. Audio Reconstruction

For speech data, we use DNS-MOS (Reddy et al., 2022), UT-MOS (Saeki et al., 2022), PESQ, STOI (Short-Time Objective Intelligibility), VISQOL (speech version), and STFT loss as metrics.

For sound and music data, we use VISQOL (audio version), STFT loss, and Mel loss. Furthermore, following (Kumar et al., 2023), we conduct the MUSHRA subjective test for speech, sound, and music. Specifically, we hire 10 audio-related researchers to conduct the MOS evaluation. We ask the listeners to rate each audio, with scores ranging from 0 to 100. Refer to D.5 for the details.

**Evaluation Datasets:** For speech data, we evaluate on a subset of VCTK (Veaux et al., 2017) (200 speech utterances) and a subset of the LibriTTS test clean set (Zen et al., 2019) (400 speech utterances). For sound data, we evaluate on a subset of the AudioCaps validation set (Kim et al., 2019) (200 sound utterances). For music data, we evaluate on a subset of the MusicCaps (Agostinelli et al., 2023) dataset (200 music utterances).

## D.2. Audio Semantic Information

Previous SSL models, such as Hubert (Hsu et al., 2021) and WavLM (Chen et al., 2022a), have shown that semantic-rich representations can be used to solve downstream recognition tasks by fine-tuning several adaptor layers. Inspired by these works, we propose evaluating the performance of the audio tokenizer for downstream recognition tasks. We use the quantized features of the audio tokenizer as the input for downstream tasks. We follow two popular benchmarks: SUPERB (Yang et al., 2021) and ARCH (La Quatra et al., 2024).

For speech data, we conduct the automatic speech recognition (ASR) task on the LibriSpeech (Panayotov et al., 2015) dataset and the emotion classification (EC) task on the EMOVO (Costantini et al., 2014) dataset. For the ASR task, we train on the LibriSpeech train-100 set and evaluate on the LibriSpeech test clean set. For the EC task, we follow ARCH (La Quatra et al., 2024) to split the training and test sets.

For sound data, we conduct the sound classification task on the ESC-50 dataset (Piczak, 2015). For music data, we conduct the music classification task on the Medley-Solos-DB dataset (Lostanlen & Cella, 2016). For both tasks, we follow the ARCH benchmarking settings to split the training and test sets.

For all experiments, we train for 10 epochs with the same learning rate and batch size. For the automatic speech recognition

task, we use word error rate (WER) as the metric. For the other classification tasks, we use accuracy as the metric.

### D.3. LM-based Audio Understanding

**Overview** To further validate whether the audio tokenizer is suitable for building an audio language model, we propose conducting an audio understanding task using discrete tokens as input. We conduct three tasks: automatic speech recognition (ASR), audio captioning, and music captioning. We use the framework introduced in Section A. For audio data, we use the audio tokenizer to encode it as discrete tokens; for text data, we use the BPE tokenizer of LLAMA 3.2. We construct the sequence as [*audio token*, *text token*], then the model is asked to predict the text token based on the previous audio token.

**Training Data** For the ASR task, we select 2,000 hours of LibriHeavy speech data (Kang et al., 2024). For the audio captioning tasks, we use AudioCaps (Kim et al., 2019) and BBC sound effects (Mei et al., 2023). For the BBC sound effects, we cut off the first 10 seconds of audio if the utterance duration is greater than 10 seconds. Finally, we obtain about 500 hours of sound data. For the music captioning task, we use a subset of the Million Song dataset. We cut off the first 10 seconds of music data for each utterance, which results in about 500 hours of music data. For the corresponding captions, we use LPMusicCaps (Doh et al., 2023).

**Test Data** For the ASR task, we evaluate on the LibriSpeech test clean set. For the audio captioning task, we evaluate on the AudioCaps dataset (Kim et al., 2019). For the music captioning task, we evaluate on the MusicCaps dataset (Agostinelli et al., 2023).

**Metrics** Similarly, we use WER as the evaluation metric for the ASR task. For audio and music captioning, we follow (Drossos et al., 2020) and adopt BLEU-1, BLEU-2, BLEU-3, METEOR, ROUGE-L, CIDEr-D, SPICE, and SPIDEr metrics.

**Inference Setting** For inference, we directly use the top-k sampling strategy and set $k = 30$ for all experiments.

### D.4. LM-based Audio Generation

We also perform audio generation tasks, including text-to-speech, text-to-sound, and text-to-music generation. Similarly, we construct the sequence as [*text token*, *audio token*], then the model is asked to predict the audio token based on the previous text token.

**Training and Test Data** We use the same training and test data as the audio comprehension task.

**Metrics** For TTS evaluation, we use WER to evaluate robustness, and UTMOS and DNSMOS are used to assess speech quality. For text-to-sound and text-to-music, we follow previous works AudioGen (Kreuk et al., 2022), using Fréchet Audio Distance (FAD), Kullback-Leibler (KL) Divergence, and Fréchet Distance (FD) for audio fidelity and similarity.

**Inference Setting** During the inference stage, we use the top-k sampling strategy and set $k = 30$ for all experiments.

### D.5. Subjective Evaluations

For the subjective evaluations, we adopt the approach used in previous works (Kumar et al., 2023; Parker et al., 2024) and use the MUSHRA format without a hidden anchor. Listeners are asked to compare multiple versions of an example simultaneously, including both a labeled reference and a hidden reference. They are given the following instructions: "Please assess the quality similarity between an audio sample and its reference. Listen carefully to the reference audio, then rate the quality of each test clip in comparison. A score of 0 indicates no resemblance to the reference, while a score of 100 means it is identical to the reference." We randomly select 10 samples from each category (speech, music, and sound) in the test set, ensuring that each sample receives 10 ratings.

## E. Audio Tokenizer Baselines

To make a fair comparison, we classify the audio tokenizers into two types: (1) speech-based tokenizers, which are trained on speech datasets, and (2) audio-based tokenizers, which are trained on speech, sound, and music datasets.

### E.1. Speech Tokenizer

For speech data, we compare with:

*Table 12.* The performance comparison on LibriTTS test clean. **Bold** for the best result and underline for the second-best result.

| Models | FPS/TPS | CS/BR | Reconstruction | | | | | Efficiency | |
| --- | --- | --- | --- | --- | --- | --- | --- | --- | --- |
| | | | UTMOS (↑) | DNS-MOS (↑) | VISQOL (↑) | STOI (↑) | PESQ (↑) | Model size (M) (↓) | RTF (↓) |
| Encodec | 50/400 | 1024/6kbps | 3.30 | 3.76 | 3.95 | 0.94 | 2.72 | 14 | 0.019 |
| Encodec | 50/150 | 1024/1.5kbps | 2.02 | 3.27 | 3.83 | 0.88 | 1.79 | 14 | 0.019 |
| DAC | 50/150 | 1024/1.5kbps | 2.61 | 3.36 | 3.85 | 0.89 | **1.96** | 71 | 0.026 |
| Wavtokenizer | 40/40 | 4096/0.48kbps | 3.65 | 3.61 | 3.80 | 0.87 | 1.81 | 77 | 0.017 |
| StableCodec | 25/25 | 46656/0.4kbps | **4.20** | **3.74** | 3.51 | 0.88 | 1.85 | 950 | 0.039 |
| MimiCodec (3 RVQ) | 12.5/37.5 | 2048/0.41kbps | 2.82 | 3.28 | 3.34 | 0.83 | 1.40 | 75.6 | 0.023 |
| ALMTokenizer (Ours) | 12.5/37.5 | 2048/0.41kbps | 3.68 | 3.64 | **3.90** | **0.90** | 1.92 | 174 | 0.031 |

(1) Encodec (Défossez et al., 2022), a SOTA audio codec model trained on large-scale speech, sound, and music datasets. The official open-sourced 24 kHz version is used.

(2) DAC-Codec (Kumar et al., 2023), which offers very high reconstruction performance. It is trained on large-scale speech, sound, and music datasets. The official open-sourced 24 kHz version is used.

(3) MimiCodec (Défossez et al., 2024), a SOTA low-bitrate speech codec model trained on a large-scale speech dataset. The sampling rate is 24 kHz.

(4) SpeechTokenizer (Zhang et al., 2023), a semantic-rich speech codec model trained on a large-scale speech dataset. The sampling rate is 16 kHz.

(5) WavTokenizer (Ji et al., 2024), an audio codec tokenizer trained on large-scale speech, sound, and music datasets. The sampling rate is 24 kHz.

To make a fair comparison, for Encodec, DAC-Codec, and SpeechTokenizer, we use the first three RVQ layers to control the bitrate during inference.

### E.2. Audio Tokenizer

For sound and music data, we compare with Encodec, DAC-Codec, and WavTokenizer. These three models are trained on large-scale speech, sound, and music datasets.

### E.3. Semantic Models

Furthermore, to evaluate the performance of semantic information, we also introduce several SSL-based models. For speech, we use WavLM (Chen et al., 2022a) and HuBERT (Hsu et al., 2021). For sound and music, we use BEATs (Chen et al., 2022b) and Wav2Vec2-AudioSet [6].

## F. More audio tokenizer evaluation experiments

### F.1. The subjective evaluation for audio tokenizer

Table 7 shows the subjective evaluation results for audio tokenizer.

### F.2. Evaluation results on LibriTTS test clean

We report the reconstruction performance evaluated on a subset of the LibriTTS test clean set, where we randomly select 400 speech utterances. Additionally, we calculate the Real-Time Factor (RTF) and model size to assess efficiency. For RTF evaluation, we use an NVIDIA A100 GPU to evaluate all models.

### F.3. Length generalization

StableCodec (Parker et al., 2024) highlights that the introduction of transformer-based architectures can lead to the length generalization problem. For instance, the training data of ALMTokenizer consists of 5-second segments, whereas the test

---

[6]https://huggingface.co/ALM/wav2vec2-large-audioset

*Table 13.* Objective metrics for the ALMTokenizer and baselines, evaluated on utterances from length 4s to 10s, showing generalization of models across lengths

| Model | FPS | TPS | Bitrate | PESQ (↑) | UT-MOS (↑) | VISQOL (↑) | STOI (↑) | DNSMOS (↑) |
|-------|-----|-----|---------|----------|------------|------------|----------|------------|
| **4 seconds** | | | | | | | | |
| Encodec | 50 | 150 | 1.5kbps | 1.97 | 2.64 | 3.62 | 0.80 | 3.26 |
| DAC | 50 | 150 | 1.5kbps | 2.1 | 3.17 | 3.65 | 0.81 | 3.26 |
| Ours | 12.5 | 37.5 | 0.41kbps | 1.84 | 3.63 | 3.69 | 0.79 | 3.41 |
| **6 seconds** | | | | | | | | |
| Encodec | 50 | 150 | 1.5kbps | 1.97 | 2.54 | 3.63 | 0.81 | 3.26 |
| DAC | 50 | 150 | 1.5kbps | 2.0 | 3.11 | 3.65 | 0.81 | 3.28 |
| Ours | 12.5 | 37.5 | 0.41kbps | 1.89 | 3.66 | 3.75 | 0.81 | 3.62 |
| **8 seconds** | | | | | | | | |
| Encodec | 50 | 150 | 1.5kbps | 1.96 | 2.52 | 3.63 | 0.81 | 3.34 |
| DAC | 50 | 150 | 1.5kbps | 2.1 | 3.18 | 3.66 | 0.81 | 3.28 |
| Ours | 12.5 | 37.5 | 0.41kbps | 1.95 | 3.55 | 3.74 | 0.81 | 3.66 |
| **10 seconds** | | | | | | | | |
| Encodec | 50 | 150 | 1.5kbps | 1.95 | 2.53 | 3.65 | 0.81 | 3.32 |
| DAC | 50 | 150 | 1.5kbps | 2.1 | 2.19 | 3.67 | 0.81 | 3.25 |
| Ours | 12.5 | 37.5 | 0.41kbps | 1.96 | 3.54 | 3.73 | 0.81 | 3.66 |

data comprises segments of varying durations. We evaluate the model across four distinct length levels: 4, 6, 8, and 10 seconds. Encodec and DAC are selected as baselines due to their reliance on convolutional layers, which demonstrate robustness to variable input lengths. As shown in Table 13, the evaluation results indicate that ALMTokenizer effectively handles inference across these diverse lengths. These findings suggest that ALMTokenizer exhibits strong generalization capabilities with respect to input length variation.

### F.4. Compared to diffusion-based audio codec models

We compare ALMTokenizer with an alternative family of audio tokenizers that leverage discrete semantic tokens derived from self-supervised pre-trained (SSL) models (e.g., Hubert (Hsu et al., 2021), WavLM (Chen et al., 2022a), AudioMAE (Huang et al., 2022)). These models first quantize the SSL features into semantic tokens and subsequently use a generative model to resynthesize the waveform. Diffusion (Ho et al., 2020) and Flow-Matching (Lipman et al., 2022) are two popular generative models. Previous works, such as GLM4-Voice tokenizer (Zeng et al., 2024) and SemantiCodec (Liu et al., 2024), have demonstrated success using diffusion-based decoders. However, such strategies tend to result in significant information loss. For instance, the semantic tokens in GLM4-Voice lack timbre information and require additional prompts to control timbre during decoding. Notably, the open-sourced GLM4-Voice tokenizer uses a fixed timbre, meaning that any speech encoded by GLM4-Voice will lose its original timbre. To address this information loss in semantic tokens, SemantiCodec introduces acoustic streaming to enhance waveform reconstruction. A key concern, however, is that both SemantiCodec and GLM4-Voice tokenizers demand significantly more computational resources during the inference stage. In the following, we present a comprehensive comparison between ALMTokenizer and SemantiCodec, focusing on the following aspects: (1) reconstruction performance for speech, sound, and music; (2) semantic information performance for speech, sound, and music; and (3) computational resource requirements during inference, measured using RTF.

Table 14 shows the speech reconstruction and semantic performance, where we observe that ALMTokenizer outperforms the alternatives in both aspects while using less bitrate. Table 15 presents experimental results for sound and music data, where ALMTokenizer again demonstrates superior performance across all metrics compared to SemantiCodec. In Table 16, we present the model size and RTF metrics, showing that ALMTokenizer has fewer model parameters and significantly surpasses SemantiCodec in inference speed (0.031 vs 0.92).

## G. The details of ALMTokenizer structure and training

### G.1. Model structure

Table 17 gives the details of ALMTokenizer configuration, which results in 174M parameters. In all of experiments, for the MAE-transformer encoded and decoder, we adopt a 8 layer transformer layers.

*Table 14.* The performance comparison between ALMTokenizer and SemanticCodec on VCTK dataset.

| Models | FPS/TPS | CS/BR | Reconstruction | | | | | Semantic | |
| | | | UTMOS (↑) | DNS-MOS (↑) | VISQOL (↑) | STOI (↑) | PESQ (↑) | ASR (↓) | EC (↑) |
| --- | --- | --- | --- | --- | --- | --- | --- | --- | --- |
| SemantiCodec | 50/50 | 16384/0.68kbps | 3.2 | 3.57 | **3.90** | 0.81 | 1.76 | 48.3 | 17.8 |
| ALMTokenizer | **12.5/37.5** | **2048/0.41kbps** | **3.76** | **3.64** | 3.78 | **0.81** | **2.0** | **18.3** | **29.0** |

*Table 15.* The performance comparison between ALMTokenizer and SemanticCodec on Music (MusicCaps) and sound data (AudioCaps).

| Models | FPS/TPS | CS/BR | Reconstruction | | | Semantic |
| | | | Mel loss (↓) | STFT loss (↓) | VISQOL (↑) | Classification (↑) |
| --- | --- | --- | --- | --- | --- | --- |
| **Sound data** | | | | | | |
| SemantiCodec | 50/50 | 16384/0.68kbps | 18.45 | 1.40 | 2.47 | 38.8% |
| ALMTokenizer | **12.5/37.5** | **2048/0.41kbps** | **15.0** | **1.24** | **2.99** | **44%** |
| **Music data** | | | | | | |
| SemantiCodec | 50/50 | 16384/0.68kbps | 47.9 | 1.58 | 2.49 | 48% |
| ALMTokenizer | **12.5/37.5** | **2048/0.41kbps** | **34.4** | **1.32** | **3.96** | **59%** |

**Patchify and UnPatchify modules** A single-channel audio signal $x \in \mathcal{R}^{1 \times N}$ (where $N$ denotes the sampling points) is processed through the Encodec-style Patchify and UnPatchify modules, which adopt the same structure as Encodec (Défossez et al., 2022), consisting of four convolutional blocks. Each convolutional block consists of a residual unit followed by a down-sampling layer. These convolution blocks effectively encode the audio signal $x$ into an audio frame representation $e \in \mathcal{R}^{T \times d}$, where $T$ denotes the number of frames and $d$ denotes the dimension of each vector. The convolution blocks are followed by a two-layer LSTM for sequence modeling, followed by a final 1D convolutional layer with a kernel size of 7 and $D$ output channels. The UnPatchify module mirrors the Patchify architecture by substituting stride convolutions with transposed convolutions and reversing the stride order.

For the StableCodec-style Patchify and UnPatchify modules, we follow the approach in StableCodec (Parker et al., 2024) and use a reshape operation to transform $x \in \mathcal{R}^{t \times sr}$ into $e \in \mathcal{R}^{T \times d}$, where $T = N/320$ and $d = 320$. We then apply a linear layer to map the dimension to $D$. Finally, we add four transformer layers for sequence modeling. Similarly, the UnPatchify module mirrors the Patchify architecture.

**Discriminators** For the discriminators, we follow prior work (Défossez et al., 2022), which combines mel-spectrogram and log-mel-spectrogram features and inputs them into a network consisting of several convolutional layers. Specifically, we use six discriminators with different configurations: the hidden dimensions are set as 64, 128, 256, 512, 512, 512, and the hop lengths are set as 32, 64, 128, 256, 512, 1024.

### G.2. Reconstruction loss and adversarial loss for ALMTokenizer

Let the reconstructed signal be $\hat{x}$. For the reconstruction loss, we design it from two perspectives: the time domain and the frequency domain. We first compute the $L_1$ loss between $x$ and $\hat{x}$ in the time domain. Next, we compute the $L_1$ loss between the STFT spectrogram of $x$ and $\hat{x}$ in the frequency domain. Following (Wang et al., 2024b), we employ a sub-band split strategy to divide the spectrogram into several parts. The adversarial loss is employed to enhance the perceptual quality of the generated audio:

$$\mathcal{L}_d = \frac{1}{K} \sum_{i=1}^{K} max(0, 1 - D_k(x)) + max(0, 1 + D_k(\hat{x})) \tag{4}$$

where $K$ denotes the number of discriminators. During the training stage, the adversarial loss for the generator is computed as a hinge loss over the logits of these discriminators:

$$\mathcal{L}_{adv} = \frac{1}{K} \sum_{i=1}^{K} max(0, 1 - D_k(\hat{x})) \tag{5}$$

The feature loss $\mathcal{L}_{feat}$ is computed by taking the average absolute difference between the discriminator's internal layer outputs for the generated audio and those for the corresponding real audio.

*Table 16.* The model size and RTF comparison between SemantiCodec and ALMTokenizer.

| Model | Model size (M) ($\downarrow$) | RTF ($\downarrow$) |
|---|---|---|
| SemantiCodec | 507 | 0.92 |
| ALMTokenizer (Ours) | **174** | **0.031** |

| | ALMTokenizer |
|---|---|
| Input shape | (B, 1, N) |
| Patchify module (output) | (B, T, d), T=N/320 |
| Token Interleaving and Retrieval | $w \in [2, 3, 4, 5, 6, 7, 8, 9, 10]$ |
| Dimension of transformer encoder | 256 |
| The number of transformer encoder | 24 |
| Dimension of transformer decoder | 512 |
| The number of transformer decoder | 24 |
| Codebook size | 2048 |
| VQ layers | 3 |
| Number of Transformer heads | 64 |
| UnPatchify module (output) | (B, 1, N) |

*Table 17.* ALMTokenizer model backbone configurations

## G.3. Training details

The AdamW optimizer is used in the training. We set the learn rate as $1e-4$. We train the model with 200k steps. The final loss as following shows. We set $\lambda_1 = 0.5$ and $\lambda_2 = 0.1$ during our experiments. We conduct all of the experiments with 4 NVIDIA A100-80G GPUs.

$$\mathcal{L} = \mathbf{L}_{adv} + \mathbf{L}_{feat} + \mathbf{L}_{rec} + \lambda_1 \mathbf{L}_{MAE} + \lambda_2 \mathbf{L}_{AR} \qquad (6)$$

# H. Reproducibility Statement

To enhance reproducibility, we provide the pseudocode of ALMTokenizer. In the future, we plan to improve both the model structure and training data to obtain more robust models, especially for music and sound, and release the code for the research community.

*Listing 1.* Pseudocode of ALMTokenizer

```
class ALMTokenizer:
    def __init__(
        self,
        transformer_encoder_args,
        transformer_decoder_args,
        mae_decoder_args,
        depth_gpt_args,
        patchify_args,
        encoder_embed_dim,
        decoder_embed_dim,
        semantic_prior_path,
        mask_rate,
        window_sizes = [2,3,4,5,6,7,8,9,10],
    ):
        self.window_sizes = window_sizes
        self.transformer_encoder = Transformer(transformer_encoder_args)
        self.transformer_decoder = Transformer(transformer_decoder_args)
        self.mae_decoder = Transformer(mae_decoder_args)
        self.Patchify = Encodec_encoder(patchify_args)
        self.UnPatchify = Encodec_decoder(patchify_args)
```

```python
        self.cls_token = nn.Parameter(torch.zeros(1, 1, encoder_embed_dim))
        self.mask_token =  nn.Parameter(torch.zeros(1, 1, decoder_embed_dim))
        checkpoint = torch.load(semantic_prior_path, map_location="cpu")
        self.vq = RVQ_semantic(
            input_dim=encoder_embed_dim,
            semantic_prior = checkpoint,
            layers = 3)
        self.depth_gpt = GPT_decoder(depth_gpt_args)
        self.tmp_window_size = 6
        self.mask_rate = mask_rate

    def Encoder_token_Interleaving(self, x):
        B, T, D = x.shape  # batch, length, dim
        cls_tokens = self.cls_token.repeat(B, (T//self.tmp_window_size), 1).unsqueze(2)
        new_T = T + (T // self.tmp_window_size)
        x_reshaped = x.reshape(B, T // self.tmp_window_size, self.tmp_window_size, D)
        x_with_cls = torch.cat([x_reshaped, cls_tokens], dim=2)
        new_x = x_with_cls.reshape(B, -1, D)
        return new_x

    def Encoder_token_Retrieval(self, x):
        B, new_T, D = x.shape
        original_T = new_T - new_T // (self.tmp_window_size + 1)
        mask_indices = [(i + 1) * (self.tmp_window_size + 1) - 1 for i in range(original_T
            // self.tmp_window_size)]
        cls_tokens = new_x[:, mask_indices, :]
        return cls_tokens

    def Decoder_token_Interleaving(self, en_token):
        B, T, D = en_token.shape
        x = self.mask_token.repeat(B, 1, 1)
        new_T = en_token.shape[1]*self.tmp_window_size + en_token.shape[1]
        x = x.repeat(1, en_token.shape[1]*self.tmp_window_size, 1)
        x = x.reshape(B, -1, self.tmp_window_size, D)
        x_with_masks = torch.cat([x, en_token.unsqueeze(2)], dim=2)
        new_x = x_with_masks.reshape(B, -1, D)
        return new_x

    def Decoder_token_Retrieval(self, new_x):
        B, new_T, D = new_x.shape
        num_masks = new_T // (self.interval + 1)
        original_T = new_T - num_masks
        mask_indices = [(i + 1) * (self.interval + 1) - 1 for i in range(num_masks)]
        all_indices = list(range(new_T))
        mask_indices = [i for i in all_indices if i not in mask_indices]
        mask_frames = new_x[:, mask_indices, :]
        return mask_frames

    def forward(
        self,
        x,
    ):
        x_len = x.shape[-1]
        self.tmp_window_size = choice(self.window_sizes)
        emb_frames = self.Patchify(x)
        if self.training:
            emb_frames_mask = self.apply_mask(emb_frames, mask_rate = self.mask_rate)
        interleving_frames = self.Encoder_token_Interleaving(emb_frames_mask)
        predict_latent = self.mae_decoder(interleving_frames)
        mae_loss = L1_loss(predict_latent, emb_frames)
        latent_tokens = self.transformer_encoder(interleving_frames)
        query_token = self.Encoder_token_Retrieval(latent_tokens)
        Quantized_token, codes, all_quantized = self.vq(query_token)
        cat_quantized = []
        for q_emb in all_quantized:
```

```
        q_emb = q_emb.reshape(-1, q_emb.shape[-1]).unsqueeze(1)
        cat_quantized.append(q_emb)
    cat_quantized = torch.cat(cat_quantized, dim=1)
    gpt_loss = self.depth_gpt.compute_prior_loss(cat_quantized)
    de_interleving_frames = self.Decoder_token_Interleaving(Quantized_token)
    de_latent_token = self.transformer_decoder(de_interleving_frames)
    mask_tokens = self.Decoder_token_Retrieval(de_latent_token)
    x_ = self.UnPatchify(mask_tokens)
    return x_, mae_loss, gpt_loss
```

## I. Limitation

In this study, we present ALMTokenizer, a low-bitrate, semantic-rich audio codec tokenizer. We demonstrate that ALM-Tokenizer excels in both reconstruction and semantic information retention under low-bitrate conditions. However, we acknowledge that there is still significant room for improvement in reconstruction performance, particularly for sound and music data. Building an audio tokenizer for sound and music in the low-bitrate setting poses additional challenges. In terms of semantic information, ALMTokenizer still lags behind traditional SSL models. Although we propose several training losses to enhance semantic information in the codec model, the improvements are limited and, in some cases, negatively impact reconstruction quality. We recognize the need for a careful design and balance of these semantic loss terms. Additionally, the multi-stage training strategy increases training complexity. These training strategy brings waste. Most of the components are eventually discarded, e.g. MAE-transformer encoder/decoder, MAE-decoder, and depth AR-transformer. These components would have made sense to still utilize them for some purpose, e.g. the AR decoder could have been used to initialize the depth transformer in the Language modeling task. These concerns are left for future work.

