# OpenReview forum: "ALMTokenizer: A Low-bitrate and Semantic-rich Audio Codec Tokenizer for Audio Language Modeling"
_ICML.cc/2025/Conference — ICML 2025 poster_

### Official Review · Reviewer_4P9V · 2025-03-11

**Overall Recommendation:** 4

**Summary:**

The paper introduces ALMTokenizer, a low-bitrate and semantically rich audio codec.

It incorporates a novel fixed-interval query interleaving mechanism which extracts contextual features from the acoustic features and quantizes (using RVQ) only the contextual features extracted by these queries, thus achieving low bitrates, empowered by transformer based encoder and decoder. The decoder receives the queries and the interval, and inserts mask tokens, which are then converted into acoustic features. Additionally the VQ codebook vectors are initialized from k-means clusters of wav2vec2 and BEATS and kept fixed during training to provide strong semantic priors to the quantization module.

Training involves two stages, first the Encodec style patchify/unpatchify modules are trained along with some dedicated stage one encoder/decoder transformers without a quantizer, with MAE loss. The goal of the first stage training is to enable the frontend patchify module to learn semantically rich features. Second stage intializes the patchify/unpatchify with the stage 1 parameters and patchify is kept frozen. Second stage is trained with a combination of several objective functions including MAE loss, AR loss and the standard codec reconstruction and GAN losses.

Training and evaluation is conducted on a mix of speech, music and general purpose audio datasets. Evaluation results indicate strong performance on semantic tasks while being competitive in terms of reconstruction quality.

## Update after rebuttal
After reading the rebuttal and discussions with the all reviewers, especially the additional experiments in response to reviewer btwe, I am confident that this work is valuable to the neural audio codec community. I keep my Accept recommendation.

**Claims And Evidence:**

The paper makes the following claims with regards to the codec:

- **low-bitrate**: experimental results show that it works well, competitive reconstruction performance.
- **semantic-rich**: experimental results show that it achieves good results on speech and emotion recognition, and sound classification tasks.
- **latent space optimized for AR modeling**: while the experimental result shows that the token prediction accuracy improves by using the AR loss, the actual effect of it does not seem to be too significant in terms of speech generation tasks (Fig. 3). In fact, from the ablation study in Table 6, it seems that not using the LM loss is beneficial for several metrics.

**Essential References Not Discussed:**

Most essential related works have been discussed.

**Experimental Designs Or Analyses:**

Experimental design is mostly fine.

I have one concern, both Encodec and DAC are used at 1.5kbps, which are the lowest settings of the respective models. While this is fair in terms of the number of RVQ layers, it would have been nice to see a comparison with their full potential as well.

**Methods And Evaluation Criteria:**

Proposed methods and evaluation criteria do make sense for the application.

However, there are many training/evaluation datasets utilized for different tasks and it is sometimes difficult to follow. Would be better for all tables/figures to mention which evaluation set it used.

Also, Table 1 and Table 6, both utilize the same metrics but are performed on different evaluation datasets, so it is difficult to contextualize the ablation study with the main result.

Most of the results are reported without confidence intervals, which makes it difficult to judge whether the difference in metrics is statistically significant or not.

**Other Comments Or Suggestions:**

1.  Table 7, codebook size and FPS columns seem to be interchanged.
2. Line 216: "we found that using a large mask rate will significantly influence the reconstruction performance", for the better or worse? Is there an experimental verification of this statement?
3. I think the subjective evaluation result should be included in the main content and not relegated to the Appendix. Maybe integrating in Table 1, and instead of reporting two automatic quality metrics (UTMOS and DNSMOS), it could be one automatic and the MUSHRA score.
4. It is better to include an explanation of the wastefulness and energy efficiency of the training pipeline in a Broader Impact Statement.
5. The paper relies a lot on the Appendix, especially for the experimental sections, unless the reader carefully goes through the Appendix, several details might be missed.

**Other Strengths And Weaknesses:**

**Strengths**
- Query token interleaving is a very elegant idea and it has been demonstrated that it works well.

**Weaknesses**
1. The training strategy seems to be too wasteful. Most of the components are eventually discarded, MAE-transformer encoder/decoder in stage 1, MAE-decoder in stage 2, AR-transformer in stage 2, etc. While I understand the purpose of them, but simply discarding is very wasteful in terms of compute and energy efficiency. It would have made sense to still utilize them for some purpose, for example, stage1 MAE encoder could have been used to initialize the enocder in stage2, and stage1 MAE decoder could have been used to initialize stage 2 MAE decoder. Similarly, the AR decoder (depth GPT) could have been used to initialize the depth transformer in the Language modeling task, etc.
2. The variable bitrate strategy is useful for the model as a pure codec, but its usefulness for downstream LM tasks is not explored. The LM is trained with a fixed bitrate and I think, it cannot generalize to a different bitrate, while in the case of variable RVQ based tokenizers, the downstream model can also take advantage of the variable bitrate for tradeoffs in quality and efficiency.

**Questions For Authors:**

Please see above sections.

**Relation To Broader Scientific Literature:**

There has been significant recent efforts in creating low bitrate codecs which are easy to use for downstream language models.

While the overall structure of the proposed method takes inspiration from these related works (using Encodec style patchify/unpathify, transformers before and after RVQ), the key novelty lies in utilizing interleaved query vectors to encode contextual information and only quantizing these query vectors.

This method also enables an alternative approach to variable bitrate by changing the frequency of the query interleaving, unlike previous methods which typically use variable number of RVQ levels (SoundStream, Encodec, DAC).

**Theoretical Claims:**

There are no theoretical claims in the paper.

---

> ### Author Rebuttal · Authors · 2025-03-30
>
> We thank the reviewer for recognizing our contributions.
>
> **Q1:**  latent space optimized for AR modeling...it seems that not using the LM loss is beneficial for several metrics.
>
> **A:** We appreciate this comment. We acknowledge that introducing the autoregressive (AR) loss may slightly impact reconstruction metrics. As discussed in the Limitations section, we emphasize this trade-off to encourage future research on achieving a better balance between reconstruction quality and modeling efficiency.
>
>  **Q2:** Methods And Evaluation Criteria:  However, there are many training/evaluation datasets utilized for different tasks ... mention which evaluation set it used ...
>
> **A:** We appreciate this comment. We will update our paper to explicitly specify the evaluation set used for all tables and figures in the final version. Furthermore, both Table 1 and Table 6 report results on the VCTK dataset. In Appendix Table 12, we present a reconstruction performance comparison between our proposed method and previous works on the LibriTTS test set. Additionally, we have reported results with 95\% confidence intervals for subjective evaluations. For objective reconstruction metrics, since they are deterministic, confidence intervals were not previously included. However, for generation experiments, we will incorporate confidence intervals by sampling multiple times in the final version.
>
> **Q3:** I have one concern, both Encodec and DAC are used at 1.5kbps...potential as well.
>
> **A:**  We appreciate this comment. In line with MimiCodec, we will include results for Encodec and DAC in two configurations: (1) using 3 RVQ layers, as reported in our paper, and (2) using 8 RVQ layers to demonstrate their full potential.
>
> **Q4:** The training strategy seems to be too wasteful... initialize the depth transformer...
>
> **A:** We appreciate this constructive comment, which provides valuable insights for improving our work. In particular, leveraging the AR decoder to initialize the deep transformer in language modeling is an interesting idea. We find this direction highly promising and plan to explore it in future work. Additionally, we will incorporate this discussion into the final version of our paper to inspire further research in this area.
>
> **Q5:** The variable bitrate strategy is useful for the model as a pure codec...
>
> **A:** We appreciate and agree with the reviewer that the applicability of the variable bitrate strategy for downstream language modeling tasks remains unexplored. Our proposed method primarily facilitates the selection of codec models with different frame rates, providing greater flexibility in bitrate allocation. We will add this discussion into the Limitation part.
>
> **Q6:** Table 7, codebook size and FPS columns seem to be interchanged.
>
> **A:** Thank you for your help to find this mistake. We will update it in the final version.
>
> **Q7:** Line 216: "we found ...of this statement?
>
> **A:**  In our experiments, we observed that a high masking rate negatively impacts reconstruction performance. We evaluated three masking rate ranges: 10–20\%, 20–30\%, and 30–40\%. As shown following, higher masking rates (30–40\%) improve semantic representation but degrade reconstruction quality. Based on these findings, we adopt an intermediate masking range of 20–30\% to balance semantic preservation and reconstruction fidelity.
>
> | mask rate range | UTMOS  | DNSMOS  | VISQOL | PESQ | STOI  | ASR   | ER   |
> |-----------------|--------|---------|--------|------|-------|-------|------|
> | 10-20%          | 3.77   | 3.62    | 3.80   | 2.0  | 0.81  | 18.7  | 27.7 |
> | 20-30%          | 3.76   | 3.64    | 3.78   | 2.0  | 0.81  | 18.3  | 29.0 |
> | 30-40%          | 3.36   | 3.06    | 3.31   | 1.58 | 0.77  | 18.1  | 29.6 |
>
> **Q8:** I think the subjective evaluation ... it could be one automatic and the MUSHRA score.
>
> **A:** We appreciate and agree with the reviewer that subjective evaluation performance should be presented as the primary result in the paper. Accordingly, we will update our manuscript to include the MUSHRA score results in the main text.
>
> **Q9:** It is better to include an explanation of the wastefulness...
>
> **A:** As we discussed in Q4, we will incorporate this constructive discussion into our final version, stating the existing wastefulness and listing the potential solutions.
>
> **Q10:** The paper relies a lot on the Appendix...several details might be missed.
>
> **A:** We appreciate this comment. Since the final version of ICML allows an additional page in the main text, we will move more experiments from the Appendix into the main text, such as (1) Table 10 (the subjective evaluation results) and (2) Table 11, the LM-based sound and music understanding and generation results.

---

> > ### Comment · Reviewer_4P9V · 2025-04-02
> >
> > I really appreciate the comments by the authors. It will be good to include the above mentioned changes in final version. I have no further questions and will keep my positive rating of the paper.

---

> > > ### Author Response · Authors · 2025-04-04
> > >
> > > We sincerely thank the reviewer for the feedback. We appreciate the positive view from the reviewer and are glad that all of concerns have been addressed. All of mentioned changes will be updated in the final version.
> > >
> > > Best wishes

---

### Official Review · Reviewer_THJp · 2025-03-11

**Overall Recommendation:** 4

**Summary:**

This paper introduces ALMTokenizer, a codec for speech, music and sound, which incorporates semantic and acoustic information into a single hierarchy of residual tokens with remarkable performance at a very low bitrate. The proposed improvement over previous codec include both architectural changes and training tricks (MAE, AR loss) that can be combined or used separately. The really remarkable scope of experiments and the many ideas of various importance introduced in the paper make it such that neural codec researchers will likely read it several times. Thus, I recommend an accept.

**Claims And Evidence:**

Pros:
- Claims are convincingly supported, at the exception of the one below. Overall, the experimental design is remarkably ambitious and represents the most extensive codec evaluation I have read so far.
Cons:
- One of the claims is that the "semantic priors" avoids distillation as done by previous work. The semantic prior involves training a k-means on self-supervised embeddings and using the centroids as the fixed codebook of the first VQ. This is a form of distillation, yet less costly as during training one does not need to pass audio through a teacher embedding. However, this is never compared to the distillation used by SpeechTokenizer or Mimi and the baseline in the corresponding ablation is instead a codec without distillation, which expectedly performs poorly on semantic tasks. A proper comparison should rather be done with a model using distillation.

**Essential References Not Discussed:**

None, the references are quite complete, and pretty much every open source baseline has been included in the experimental pipeline.

**Experimental Designs Or Analyses:**

Pros:
- Extensive experiments across speech, music and sounds, both in discriminative and generative settings.
- Interesting ablations and overall compelling experimental design.


Negative:
- Presentation of results: Human evaluations of audio quality are put in Appendix, while they show a worse performance for ALMTokenizer than for baselines. These results contradict claims such as "ALMTokenizer achieves better reconstruction performance at lower bitrate". Objective proxies for audio quality assessment (UTMOS, DNSMOS, MOSNet, etc.) are notoriously limited and provide a much weaker signal than actual human judgments. The MUSHRA scores should thus appear as main results of the paper, even if they depict a less positive result for the proposed model (they actually are quite good since ALMTokenizer outperforms all previous models at matching bitrates).

**Methods And Evaluation Criteria:**

See "Experimental Designs and Analyses"

**Other Comments Or Suggestions:**

1) - L230: what does "continuous" mean in that setting? that it's continuously trained? This requires more details in particular I guess a stop gradient is applied to avoid degenerate solutions? Also, does "predicting the third VQ from the first and second" mean that this small model is only autoregressive along the VQ axis or is it also autoregressive along time?.
2) L357: 12.5kHz -> 12.5Hz. Also, Mimi uses Transformers in the encoder and the decoder, is this included in this ablation? The baseline described in "The Effectiveness of Query-based Audio Compression" seems to be purely convolutional. Making sure those transformers are included would demonstrate the usefulness of the query-based proposal wrt a simple transformer.

**Other Strengths And Weaknesses:**

N/A

**Questions For Authors:**

N/A

**Relation To Broader Scientific Literature:**

ALMTokenizer introduces new ideas wrt previous work along two main axes: first, a new Transformer architecture that improves over fully convolutional codecs, and is an alternative to the Tranformers used by Mimi. Second, a set of additional losses (MAE, AR loss) to force semantic information into the learned tokens without the need for semantic distillation. Both contributions are somehow independent, and properly evaluated as such in the ablations study. Overall, the proposed methods are not groundbreaking but are conceptually sound and simple, and properly supported, such that I expect the community of neural audio codecs to build on it in the future.

**Theoretical Claims:**

No theoretical claims.

---

> ### Author Rebuttal · Authors · 2025-03-30
>
> We thank the reviewer for recognizing our contributions. We do appreciate the constructive comments the reviewer provided to us to further improve our paper. We are delighted to have the following discussion with the reviewer.
>
> **Q1:** One of the claims is that the "semantic priors" avoids distillation as done by previous work. The semantic prior involves training a k-means on self-supervised embeddings and using the centroids as the fixed codebook of the first VQ. This is a form of distillation, yet less costly as during training one does not need to pass audio through a teacher embedding...
>
> **A:** We appreciate and agree with the reviewer that semantic priors can be regarded as a form of distillation. Their advantages include: (1) reduced training cost, as they do not require audio to be passed through a teacher embedding during training; and (2) the flexibility to integrate multiple teacher models, such as Wav2Vec2 for speech semantic priors and BEATs for general sound semantic priors. In contrast, previous methods like SpeechTokenizer and MimiCodec rely on a single teacher model and primarily focus on speech semantic priors, although they can be extended to both speech and general sound.
>
> We will revise our previous claim that 'semantic priors avoid distillation' to clarify that semantic priors are indeed a form of distillation. Additionally, we will highlight their advantages and application scenarios is different with previous methods.
>
> **Q2:** Presentation of results: Human evaluations of audio quality are put in Appendix...
>
> **A:** We appreciate this comment and agree that human evaluation performance should be presented as a primary result in the paper. Accordingly, we will update our manuscript to include the MUSHRA score results in the main text.
>
> **Q3:**  L230: what does "continuous" mean in that setting? that it's continuously trained? This requires more details in particular I guess a stop gradient is applied to avoid degenerate solutions?
>
> **A:** We appreciate this comment. The term 'continuous autoregressive (AR) transformer' is used to distinguish our approach from traditional discrete AR models, which operate on discrete token sequences and are optimized using cross-entropy loss. In our study, to facilitate gradient backpropagation, we apply the AR transformer directly to continuous features (i.e., the quantized features) and optimize using mean squared error (MSE) loss. We will put these details in our final version.
>
> **Q4:** Also, does "predicting the third VQ from the first and second" mean that this small model is only autoregressive along the VQ axis or is it also autoregressive along time?.
>
> **A:** Yes, the AR model is only autoregressive along the VQ axis.
>
> **Q5:** L357: 12.5kHz -> 12.5Hz.
>
> **A:** Thank you for your help to find this mistake. We will update it in the final version.
>
> **Q6:** Also, Mimi uses Transformers in the encoder and the decoder, is this included in this ablation? The baseline described in "The Effectiveness of Query-based Audio Compression" seems to be purely convolutional. Making sure those transformers are included would demonstrate the usefulness of the query-based proposal wrt a simple transformer.
>
> **A:** We appreciate this comment. In our ablation study, The Effectiveness of Query-based Audio Compression, we compare our approach against a reproduced version of MimiCodec, which incorporates convolutional and transformer layers. The details of our reproduced MimiCodec implementation are provided in Appendix B.4. Table 7 presents the performance of reproduced MimiCodec at three different frame rates: 50 Hz, 25 Hz, and 12.5 Hz.

---

### Official Review · Reviewer_st8J · 2025-03-13

**Overall Recommendation:** 3

**Summary:**

The paper presents a method to convert an audio signal to a sequence of discrete tokens, with an aim to maximize compression (low bit rate) while retaining maximum semantic information. To achieve this goal, it introduces the use of learnable query tokens, masked auto-encoders, semantic priors (to initialize VQ layer), and AR prediction loss, in the audio tokenization pipeline. The applications include audio generation, text-to-speech and multimodal LLMs.

**Claims And Evidence:**

- Several essential concepts are not properly explained. For example, the concept of query tokens is introduced on page 3 line 157 right, without any definition. Then, line 205 left says that [CLS] is a learnable query token. Overall the concept of query token as used in this work is not clear to me.
- I have concerns regarding the novelty of the work.
  - It seems the bit-rate is reduced by hyperparameter tuning (12.5Hz, 25 Hz, 50Hz). Please correct me if I missed something essential.
  - Second, the semantic richness of tokens is achieved by query tokens, but the concept of the same is not clear to me.
  - The use of transformers, MAE, VQ layer initialization and AR prediction loss seems novel but they contribute more to the audio processing side and would be a very good contribution to audio conferences and journals.

**Essential References Not Discussed:**

Tokenization is also used for audio retrieval. One may refer to works such as "Spoken-Term Discovery using Discrete Speech Units, Interspeech 2024" and "wav2tok: Deep Sequence Tokenizer for Audio Retrieval, ICLR 2023".

**Experimental Designs Or Analyses:**

The experiments have been performed on a variety of tasks and look good.

**Methods And Evaluation Criteria:**

The paper uses a wide range of experiments and tasks to evaluate the proposed method. They aim at evaluating both compression efficiency (bit rate) and the semantic richness of the tokens. They appear alright to me.

**Other Comments Or Suggestions:**

None

**Other Strengths And Weaknesses:**

- Clarity of writing: the proposed method is not clear to me, in particular, the concept of query tokens.
- Contributions: I am not convinced about contributions to ML. They need to be highlighted by the authors.

**Questions For Authors:**

- What are query tokens? One can learn them from the data, but how are they used during inference. E.g., for text-to-speech, how do we know them; do we derive them from the given text and use for speech synthesis?
- The paper does an impressive number of experiments to show the utility of the proposed tokenizer. But the theoretical contributions to ML need to be highlighted.

**Relation To Broader Scientific Literature:**

It relates to the broader literature very well. Many state of the art tokenizers and encoders have been discussed.

**Theoretical Claims:**

As I mentioned above, several things, such as the concept of query tokens, are not clearly explained.

---

> ### Author Rebuttal · Authors · 2025-03-30
>
> We greatly appreciate the reviewer's time and patience with our paper. We are delighted to solve the concerns of reviewer one by one.
>
> **Q1:**  Several essential ... query token is not clear to me.
>
> **A:**  We appreciate your feedback regarding the clarity of the "query token" concept. Below, we provide a detailed explanation.
>
> we first revisit the workflow of previous tokenization methods, such as Encodec and SoundStream. As shown in the left part of Figure 2, the input audio data is first processed by the encoder module, transforming it into a series of audio frames, denoted as $\boldsymbol{e} \in \mathcal{R}^{T \times d}$, where $T$ represents the number of frames. The value of $T$ is determined by the down-sampling strides in the encoder module. Notably, previous works treat all audio frames equally. However, in such a setting, reducing a low frame rate (e.g., 12.5 Hz) requires increasing the down-sampling stride to 1920, which significantly degrades reconstruction due to: (1) ignoring the fact that different audio frames encode varying levels of information; and (2) failing to leverage contextual dependencies across frames.
>
> Thus, we propose a query-based compression strategy. Instead of employing large down-sampling strides, we first use a Patchify module to segment the audio into frames. Our approach then introduces learnable query tokens, which dynamically extract information across multiple frames. **Notably, query tokens are learnable embedding vectors that are updated throughout the training process.** As described in Section 3.2, these learnable query tokens are combined with the audio frames and processed by a transformer encoder, where they adaptively aggregate important information. After that the original audio frames will be ignored, and these query tokens will be used for RVQ and decoder.
>
> In summary, query tokens serve as learnable embedding vectors designed to capture holistic contextual information from the audio frames. Since the number of query tokens is smaller than the number of audio frames, which effectively reduces the frame rate while preserving essential information. The concept of query tokens is analogous to BERT [1], where a [CLS] token is placed at the beginning of a sentence to capture its semantic representation (In contrast, in our stdudy, we introduce multiple query tokens by Query token interleaving). To enhance clarity for readers, we denote this query token as [CLS] in our paper (line 205).
>
> **Q2:**   It seems the bit-rate is...
>
> **A:**  As discussed in Q1, the low bitrate is not achieved through hyperparameter tuning. Instead, we introduce query tokens that summarize contextual information across frames, where the number of query tokens directly determines the bitrate.
>
> **Q3:**  Second, the ... same is not clear to me.
>
> **A:**  We appreciate this comment. As discussed in Q1, we employ query tokens to capture holistic audio context information from audio frames. Similar to BERT, semantic information is effectively aggregated into the query tokens with the aid of a transformer encoder.
>
> **Q4:**  The use of transformers, MAE, .. seems novel ... audio conferences and journals.
>
> **A:** We thank the reviewer for recognizing our contributions. Although our approach involves audio processing techniques, its core contribution lies in machine learning methodologies—specifically, transformer-based compression and representation learning for audio modeling. Our method is not limited to audio; the query-based compression strategy can be extended to other sequential modalities. Given the increasing importance of efficient tokenization strategies in large-scale multimodal models, we believe our work is highly relevant to the ICML community.
>
> **Q4:**  Essential References Not Discussed...
>
> **A:** We appreciate and agree with the reviewer that tokenization can also be applied to audio retrieval, further highlighting the potential of our research for multimodal tasks. We are pleased to discuss the application of tokenization methods (such as DUSTED and wav2tok) in audio retrieval in our revised version.
>
> **Q5:**  Contributions: I am not convinced about contributions to ML.
>
> **A:**  Reviewer can refer to Q4.
>
> **Q6:** What are query tokens? ...for speech synthesis?
>
> **A:** The reviewer may refer to Q1 for further details. For the text-to-speech task, the corresponding query tokens can be predicted based on textual conditions. Because we use RVQ to quantize the query tokens into discrete IDs, enabling them to be modeled by an LM-based audio generation framework.
>
> **Q7:** The paper does an impressive number of experiments...
>
> **A:**  We thank the reviewer for recognizing our esperiments and contributions. The theoretical contributions to ML can refer to Q4.
>
> [1] Devlin J, et al. Bert: Pre-training of deep bidirectional transformers for language understanding. NAACL. 2019.

---

> > ### Comment · Reviewer_st8J · 2025-04-08
> >
> > Thanks for answering my questions. I appreciate the work including the extensive experiments. Please include the description of query tokens in the main paper for the benefit of a broader audience.

---

> > > ### Author Response · Authors · 2025-04-08
> > >
> > > We sincerely thank the reviewer for the feedback. We appreciate the positive view from the reviewer and are glad that all of concerns have been addressed. All of mentioned changes will be included in our final version.
> > >
> > > Best wishes

---

### Official Review · Reviewer_btwe · 2025-03-14

**Overall Recommendation:** 4

**Summary:**

The authors propose ALMTokenizer, an audio tokenizer designed to enhance compression efficiency and reconstruction quality at a low bitrate. Its key innovations include a query-based framework, semantic priors in vector quantization (VQ) codebooks by leveraging self-supervised learning (SSL) model feature clusters, MAE and LM losses, and a two-stage training approach. Experimental results show that ALMTokenizer meets or exceeds the performance of recent neural audio codecs in both compression and semantic information measures. Additionally, downstream generative modeling with ALMTokenizer’s audio tokens achieves stronger performance compared to competing tokenizers.


## update after rebuttal
After reviewing the authors’ rebuttal and their reply to the rebuttal comments, I find that the updated experimental results sufficiently address most of my concerns. I encourage the authors to incorporate the points discussed during this exchange into the revised manuscript. Given these improvements, I have updated my score from 3 to 4 and now recommend acceptance of the paper.

**Claims And Evidence:**

The proposed methods and evaluation criteria are relevant.

**Essential References Not Discussed:**

Essential references are well discussed.

**Experimental Designs Or Analyses:**

I reviewed the experimental design and analyses and found them sound.

**Methods And Evaluation Criteria:**

Proposed methods and/or evaluation criteria are reasonable and aligned with the problem/application.

**Other Comments Or Suggestions:**

I don't have any other comments or suggestions.

**Other Strengths And Weaknesses:**

Strengths:
* The proposed query-based compression framework is promising for easily adjusting compression rates, making it potentially useful across varied low-bitrate scenarios.
* The paper is well-structured, with detailed explanations of the architecture and training process. The analysis of experimental outcomes is both comprehensive and straightforward, aiding reader understanding.
*  Improved performance in downstream generative modeling including TTS indicates that the proposed approach can be integrated effectively into broader applications.

Weaknesses:
* Although the paper’s ablation study suggests that each proposed technique contributes, the simultaneous use of multiple methods (semantic prior VQ, MAE, LM loss, and two-stage training) complicates fair comparisons. For example, while semantic VQ priors provide only marginal improvements in semantic tasks, they slightly degrade audio reconstruction quality. Additionally, MAE, LM loss, and two-stage training could likely be integrated independently into other neural audio codecs. A more direct comparison using only the core techniques in the main experiments would strengthen the paper’s central claim and clarify the true necessity of the auxiliary techniques.
* The paper does not sufficiently address how changes in specific hyperparameters or components (e.g., MAE or AR loss) might affect the final model’s behavior. If the authors assert that each technique is equally important, analyzing these variations would help determine whether the approach is robust or susceptible to performance degradation under different configurations.

**Questions For Authors:**

I don't have any other questions.

**Relation To Broader Scientific Literature:**

This work contributes to improving neural audio codecs at low bit rates and demonstrates enhanced performance on downstream generative tasks. These advancements are relevant to broader audio generative modeling, including applications such as speech-language modeling, text-to-speech, and voice conversion.

**Theoretical Claims:**

I reviewed proposed methods and found them sound.

---

> ### Author Rebuttal · Authors · 2025-03-30
>
> We thank the reviewer for recognizing our contributions. We do appreciate the constructive comments the reviewer provided to us to further improve our paper. We are delighted to have the following discussion with the reviewer.
>
> **Q1** : Although the paper’s ablation study suggests that each proposed technique contributes, the simultaneous use of multiple methods (semantic prior VQ, MAE, LM loss, and two-stage training) complicates fair comparisons.
>
> **A** : We thank the reviewer for acknowledging the contributions of our proposed techniques. To systematically validate each component's effectiveness, we have conducted extensive ablation studies (Table 6 of manuscripts), including:
>
> **(1) Query-Based Framework** : We constructed a MimiCodec-style baseline [1] with identical convolutional patchify/unpatchify modules and transformer-based encoder/decoder. The key distinction lies in our proposed query-based framework, which demonstrably improves audio compression and semantic performance (Table 6, Row 3).
>
> **(2) MAE and LM (AR) Losses** : Ablation experiments confirm that both losses enhance semantic performance (Table 6, Rows 4) and Figure 3.
>
> **(3) Two-Stage Training** : Compared to one-stage training, our two-stage strategy consistently improves reconstruction and semantic metrics (Table 6, Row 6).
>
> These experiments clearly clarify the impacts of our techniques.
>
> **Q2**:  For example, while semantic VQ priors provide only marginal improvements in semantic tasks, they slightly degrade audio reconstruction quality.
>
> **A:**  We agree with the reviewer’s observation regarding the trade-off between semantic performance and reconstruction quality. One of the potential reasons is that we fix the VQ codebooks during training, which is different from the traditional VQ training (as noted in Section 3.2). This aligns with our discussion on limitations part: while semantic VQ priors improve semantic performance, optimizing the semantic information and minimize reconstruction loss remains an open challenge. We highlight this trade-off to encourage future work on balanced solutions.
>
> **Q3:**  Additionally, MAE, LM loss, and two-stage training could likely be integrated independently into other neural audio codecs.
>
> **A:** We appreciate and agree with the reviewer that our proposed technique contributes, such as MAE loss, LM loss, two-stage training strategy can be integrated into other neural audio codec models, such as Encodec and MimiCodec. As we discussed in Q1, we have conducted ablation studies to validate the effectiveness of each part. We hope these contributions will inspire broader adoption in the audio codec community.
>
> **Q4:**  The paper does not sufficiently ..., analyzing these variations would help determine whether the approach is robust or ... under different configurations.
>
> **A:** We appreciate this suggestion and provide additional analyses:
>
> **(1) Mask Rate in MAE Loss.**
> Inspired by MAE [2], we tested three group of mask rates ranges: (10–20\%), (20–30\%), and (30–40\%). The experiments as following Table shows. Results indicate that higher rates (30–40\%) benefit semantics but harm reconstruction, leading us to adopt an intermediate range (20–30\%).
>
> | mask rate range | UTMOS  | DNSMOS  | VISQOL | PESQ |  STOI |  ASR  |  ER  |
> |:---------------:|:------:|:-------:|:------:|:----:|:-----:|:-----:|:----:|
> |      10-20%     |  3.77  |   3.62  |  3.78  |  2.0 |  0.81 |  18.7 | 27.7 |
> |      20-30%     |  3.76  |  3.64   |  3.78  | 2.0  | 0.81  | 18.3  | 29.0 |
> |      30-40%     |  3.36  |   3.06  |  3.31  | 1.58 |  0.77 |  18.1 | 29.6 |
>
> **(2) The hyperparameters of MAE loss and AR loss weighting.**   To better understanding the influence of hyperparameters of MAE loss and AR loss ($\lambda_1$ and $\lambda_2$), we design 3 group settings (1, 1), (0.5, 0.5), (0.5, 0.1). The experimental results as the following Table shows. We can see that (0.5, 0.1) obtains better performance, thus we empirically choose $\lambda_1=0.5$ and $\lambda_2=0.1$ as our default setting.
>
> | $\lambda_1$  | $\lambda_2$ | UTMOS   | DNSMOS     | VISQOL    | PESQ    | STOI    | ASR    |  ER  |
> |:------:|:------:|:-------:|:----------:|:---------:|:-------:|:-------:|:------:|:----:|
> |    1   |    1   |   3.69  |    3.55    |    3.70   |   1.8   |   0.78  |  18.4  | 29.7 |
> |   0.5  |   0.5  |   3.71  |    3.58    |    3.77   |   1.9   |   0.77  |  19.0  | 28.8 |
> |   0.5  |   0.1  |   3.76  |    3.64    |   3.78    |   2.0   |  0.81   |  18.3  | 29.0 |
>
> [1] Défossez A, et al. Moshi: a speech-text foundation model for real-time dialogue[J]. 2024. \\
>
> [2] He K, et al. Masked autoencoders are scalable vision learners CVPR 2022.

---

> > ### Comment · Reviewer_btwe · 2025-04-06
> >
> > I appreciate the authors' careful responses and their efforts to address my concerns, particularly their clarification regarding one of my questions (Q4). However, it would be beneficial if the authors could explicitly highlight the advantages of their proposed main architecture compared to the baseline methods without MAE, LM loss, and two-stage training, since these components could also potentially be applied to the baselines. Nevertheless, I acknowledge and value the contributions of this study, including the introduction of these techniques, and will maintain my initial positive evaluation.

---

> > > ### Author Response · Authors · 2025-04-08
> > >
> > > We sincerely thank the reviewer for the feedback. We appreciate the positive perspective and are glad that most of the concerns have been addressed. We agree that it would be beneficial to highlight the advantages of our proposed main architecture compared to baseline methods.
> > >
> > > To demonstrate the advantages of our proposed architecture, we conducted an ablation study comparing the performance of the following two methods:
> > >
> > > (1) **The proposed query-based compression strategy**,  which uses convolutional patchify/unpatchify modules and a transformer-based encoder/decoder as the backbone, without MAE loss, LM loss, and two-stage training.
> > >
> > > (2) **The previous SOTA method, MimiCodec [1]**, using the same convolutional patchify/unpatchify modules and transformer-based encoder/decoder for a fair comparison.
> > >
> > > For the MimiCodec baseline, we applied down-sampling rates of [2, 4, 5, 6, 8], resulting in a frame rate of 12.5 Hz to match that of our proposed method. Additionally, the codebook size (2048) and number of VQ layers (3) were kept the same across both models.
> > >
> > > The experimental results are shown in the table below:
> > >
> > > |               model              |  UTMOS   | DNSMOS     | VISQOL    |  PESQ    |  STOI    |  ASR     |     ER    |
> > > |:--------------------------------:|:--------:|:----------:|:---------:|:--------:|:--------:|:--------:|:---------:|
> > > |     MimiCodec-style baseline     |   2.49   |    3.13    |   3.37    |   1.58   |   0.77   |   34.5   |   22.6   |
> > > | Proposed Query-based compression | **3.54** |  **3.41**  | **3.44** | **1.69** | **0.78** | **27.2** | **24.5** |
> > >
> > > These results clearly demonstrate the effectiveness of the proposed query-based compression strategy.
> > >
> > > Finally, we appreciate the reviewer’s suggestions to further improve our work, and we will incorporate this discussion in the final version of the paper.
> > >
> > > [1] Défossez A, et al. Moshi: a speech-text foundation model for real-time dialogue[J]. 2024.

---

### Decision · Program_Chairs · 2025-05-01

**Decision:**

Accept (poster)

**Comment:**

Summary:
The paper proposes ALMTokenizer, a novel audio codec tokenizer that advances audio language modeling by addressing two key challenges: achieving low bitrate compression while preserving semantic richness in audio representations. Existing approaches either lose semantic detail or sacrifice compression efficiency. ALMTokenizer introduces:
- A query-based compression strategy that models cross-frame audio context.
- A combination of MAE loss, VQ initialization, and AR prediction loss to improve semantic representation and latent space usability.
- A two-stage training procedure that separates semantic learning and quantization optimization.
- Evaluations that span speech, sound, and music tasks across reconstruction, understanding, and generation, with SoTA performance.

Strength:
- Innovation: The query-based compression mechanism is novel, effectively combining transformer-based context modeling with discrete token generation at low bitrates.
- Experiments: Strong results across speech, sound, and music datasets on several benchmarks, for several tasks.
- Ablation Studies: Systematic evaluation of each component in the overall approach, offering insights into what drives overall performance.
- Versatility: Demonstrates success across modalities (speech, music, sound) and tasks (understanding + generation).

Weakness:
- Complexity: While modular, the need to implement multiple components may hinder adoption of this framework.
- Gap in Performance: ASR results lag behind SSL-based models, suggesting area for improvements of this approach.

Based on reviewer feedback, discussion and AC assessment, this paper is recommended for Accept.